# Improving birth preparedness and complication readiness in rural India through an integrated microfinance and health literacy programme: evidence from a quasi-experimental study

Danish Ahmad  ,[1,2,3] Itismita Mohanty,[1] Theophile Niyonsenga  [1]

¹Health Research Institute, Faculty of Health, University of Canberra, Canberra, Australian Capital Territory, Australia
²Indian Institute of Public Health Gandhinagar (IIPH-G), Gandhinagar, Gujarat, India
³Public Health Foundation of India, New Delhi, India

**Correspondence to**
Dr Danish Ahmad;
drdanish.research@gmail.com

## ABSTRACT

**Objective** Recently, a novel community health programme—the integrated microfinance and health literacy (IMFHL) programme was implemented through microfinance-based women's only self-help groups (SHGs) in India to promote birth preparedness and complication readiness (BPCR) to improve maternal health. The study evaluated the impact of the IMFHL programme on BPCR practice by women in one of India's poorest states—Uttar Pradesh—adjusting for the community, household and individual variables. The paper also examined for any diffusion of knowledge of BPCR from SHG members receiving the health literacy intervention to non-members in programme villages.

**Design** Quasi-experimental study using cross-sectional survey data.

**Settings** Secondary survey data from the IMFHL programme were used.

**Participants** Survey data were collected from 17 244 women in households with SHG member and non-member households in rural India.

**Primary outcomes** Multivariable logistic regression was used to estimate main and adjusted IMFHL programme effects on maternal BPCR practice in their last pregnancy.

**Results** Membership in SHGs alone is positively associated with BPCR practice, with 17% higher odds (OR=1.17, 95% CI 1.07 to 1.29, p<0.01) of these women practising BPCR compared with women in villages without the programmes. Furthermore, the odds of practising complete BPCR increase to almost 50% (OR=1.48, 95% CI 1.35 to 1.63, p<0.01) when a maternal health literacy component is added to the SHGs. A diffusion effect was found for BPCR practice from SHG members to non-members when the health literacy component was integrated into the SHG model.

**Conclusions** The results suggest that SHG membership exerts a positive impact on planned health behaviour and a diffusion effect of BPCR practice from members to non-members when SHGs are enriched with a health literacy component. The study provides evidence to guide the implementation of community health programmes seeking to promote BPCR practise in low resource settings.

### Study strengths and limitations of this study

► This is the first study to evaluate the potential of delivering maternal health literacy for pregnancy complications at scale through women-based microfinance groups to rural low-income communities where higher maternal health inequities persist.

► The study's evaluation framework and research question adapted the widely used three delay model and the established social determinants of health framework.

► The evaluation design used a novel methodology to identify the programme's main effect and adjusted effects using a comprehensive list of variables using a multivariate stepwise forward selection process to examine how program-associated outcomes are influenced.

► The study was limited by a lack of randomisation in the selection of the survey blocks during programme implementation owing to pragmatic operational considerations in a high-population and low-resource setting.

## INTRODUCTION

Improving maternal health is a global priority in the attainment of the Sustainable Development Goals (SDGs). Overall, health system reforms and socioeconomic improvements in low-resource countries led to an almost 50% reduction in maternal deaths by the end of the Millennium Development Goals (MDGs) in 2015 compared with the 1990s.[1–3] However, the rate of decline was inadequate to meet the global MDG target of a 75% reduction in the maternal mortality rate (MMR).[3 4] Reflecting this, more than 295 000 maternal deaths occurred in 2017, mostly due to preventable causes across the maternity period, including hypertensive disorders of pregnancy, severe bleeding, obstructed labour and sepsis,[4–6] all conditions with established treatment options. The high number of avoidable

deaths stemming from these causes reflects the residual gap between the MDG target and available means to achieve it. The SDG has a further stringent target of reducing the MMR to 70 per 100 000 live births by 2030 for all countries.[5] However, the current situation has been further challenged by severe health service disruptions due to COVID 19. Achieving the SDGs' maternal health target requires concerted action in select high poverty and low resource countries in Sub-Saharan Africa and South Asia, where almost 99% of global preventable maternal deaths occur.[1 7 8] India has the world's second highest number of maternal deaths, with almost 45 000 maternal deaths annually.[9 10] Although India invested substantially in improving health service delivery for its predominantly rural populations, high maternal deaths remain a challenge.[6 8] Attaining the SDG target requires specific improvements in India to accelerate maternal mortality decline beyond that observed in the MDG era and the adoption of novel strategies that bridge health system inadequacies with civil society actions.[11 12]

While India has prioritised institutional delivery through a national conditional cash transfer scheme incentivising deliveries in a health facility, substantial gaps persist in early identification and care of maternal complications, especially in the antenatal and postnatal period for rural populations.[10 13 14] Inadequate referral services and limited emergency obstetric care place women at higher risk of adverse maternal outcomes in rural areas.[12–14] Health system inadequacies are further compounded by community-related delays in seeking care and reaching health facilities due to financial barriers and lack of birth and complication readiness.[13 15–19] Maternal death reviews have shown that high poverty and low development states in the north of India, especially Uttar Pradesh (UP), account for the majority of maternal deaths.[8 10 20]

Studies from India and rural UP have shown that, despite higher institutional deliveries, deficiencies in antenatal care (ANC) limit the transmission of key maternal health information such as birth preparedness and complication readiness (BPCR) to pregnant women.[14 21] Improving BPCR is strategic to reduce maternal mortality through increasing awareness and preparedness around emergency obstetrics care.[13] Pregnant women and their households are expected to receive BPCR advice as part of ANC services provided by health workers.[13] In a rural setting, irrespective of risk, where antenatal and referral services are inadequate to ensure monitoring of pregnancies, households' adoption of BPCR assumes greater importance.[14 21 22]

Additionally, studies have shown that BPCR is challenging to achieve in low-resource settings, where health system services are lacking, and households have low education and high poverty levels.[2 12 23–27] Recently, a promising community-based intervention that builds on the concept of empowering poor rural pregnant women by providing them with health literacy and finance-based assistance through women-only microfinance groups was implemented in rural UP. The integrated microfinance and health literacy (IMFHL) programme was built on other community-based programmes that have shown an improvement in routine indicators of maternal health[28–31] implemented on the microfinance platform to provide pregnant and recently delivered women with health messages. Information is provided to improve knowledge of dangerous signs related to pregnancy, childbirth and postdelivery complications and to adopt BPCR practice in order to reduce delays associated with seeking care. Microfinance-based self-help group (SHG) platforms in UP and other low-development regions of India have previously shown that integrated health literacy and SHG interventions can improve routine maternal health practices such as ANC utilisation, increase institutional deliveries and also improve newborn health practises at home, such as cord care and timely breastfeeding.[32 33] However, the impact of membership in a SHG-based health programme on the practice of BPCR among SHG members to reduce maternal mortality has not been evaluated. Also, the potential of the programme to diffuse health information from members to non-members in villages where SHGs are established is untested.

This research builds on limited evidence on the effectiveness of membership in microfinance programmes integrated with a health literacy component on BPCR practice among women in rural areas.[32–36] It is hypothesised that women in low-income rural households receiving financial access and additional health information through SHGs would reduce healthcare-seeking delays during maternal complications.

This study evaluated the impact of membership in an IMFHL programme on the practice of BPCR by women during their last pregnancy in rural UP. It also evaluated whether the IMFHL programme contributed to the adoption of BPCR among women who did not participate in the IMFHL programme as SHG members through a process of diffusion of behaviours from SHG members to neighbouring non-member households in the same villages where the programme was implemented.

## METHODS

The study used cross-sectional data from a quasi-experimental survey design, collected as part of the IMFHL programme in two rounds (pre–post programme implementation), capturing the programme's characteristics and other factors in the first round in 2015, followed by a second round of data collection 2 years into the programme in 2017.[32 33]

### Description of the IMFHL programme intervention

The IMFHL programme was implemented over 5 years (2012–2017) in two phases: an initial trial or learning phase in limited areas from 2012 to 2015, followed by at-scale implementation from 2015 to 2017.[32 33] The programme developed the final implementation design and scale-up strategy based on experiences in the pilot phase. At scale, programme implementation occurred from 2015 to 2017

in UP state, and the survey data collected at the start and end of this period are used for this paper.[32 33]

The maternal and neonatal health literacy component was layered on a pre-existing microfinance-based SHG platform towards 2015 to provide pregnant and recently delivered women with health messages in rural UP. A detailed description of the IMFHL programme context, selection of intervention, comparison and control blocks is available elsewhere.[32 33]

A woman was classified as an SHG member if she was a member herself or someone else in the household, usually a mother-in-law or sister-in-law, was a member.[32] These SHG members represented the first tier of beneficiaries who directly received health information, encouraging them to adopt desired health behaviours to promote maternal and neonatal health. As members and non-members lived in the same villages, it was expected that members receiving health information would communicate knowledge to neighbouring non-member households through informal communication networks that would gradually lead to the adoption of desired behaviours, as observed elsewhere in previous social network analysis studies.[36 37] The process of this social transfer of health information and practices through informal networking to non-members is known as the diffusion effect, also evaluated in this paper.[32 36 37] An underlying process of collective socialisation[32 37] is expected to explain the diffusion process, in which health literate SHG members serve as role models to help non-members in their community adopt protective behaviours around pregnancy and childbirth for safer maternal outcomes. This diffusion of knowledge from members (tier I) to neighbouring non-members (tier II) is depicted in figure 1.

### Study setting and participants
#### Sampling procedure
The survey sampling design followed UP's administrative hierarchy and collected data from households in *gram panchayats* (GPs) or villages from 70 blocks in 20 districts, aiming for a representative sample from the programme's coverage area.[32 38] Moreover, while the same GPs, blocks and districts were visited in both survey rounds, different households were sampled and interviewed in each round. The sampling used a multistage stratified sampling design to select blocks, GPs and households across three areas based on the IMFHL programme's exposure: intervention (SHG plus health implemented), comparator (SHG only implemented) and pure control (neither SHG nor

health implemented) areas. A detailed description of the selection strategy is available elsewhere.[32 33]

### Sample size and data
The analytical sample used in this study comprised a total of 17 244 eligible women, of whom 59% (10 097 women) did not belong to SHGs. The data were collected at the individual, household and community levels using separate structured questionnaires. The survey collected data from currently married women aged 15–49 years who had delivered an infant in the 12 months preceding the survey and from the household head and village representatives.[32 33] Thus, eligible women interviewed in round 1 (2015) would have delivered in the 12 months before the programme's start, while those women interviewed in round 2 (2017) would have delivered after the health intervention was started. The total sampling size was estimated with a 85% power, the usual 5% level of significance and a design effect of 2. The detailed sample size determination is reported elsewhere.[32 33]

### Outcome, exposure and confounding variables
#### Outcome variable
The BPCR practice captured eligible women's self-reported BPCR steps taken during the last pregnancy that would have occurred 12 months prior to the survey round(s). The outcome variable, a binary variable, was constructed with '0' representing partial or no BPCR preparation and '1' representing complete BPCR, defined as households that practised all eight BPCR steps during the last pregnancy as outlined by the World Health Organisation (WHO). In both survey rounds, eligible women, independent of the place of delivery, were asked to recall multiple responses to the question: '*What advance preparation did you/your family members make to manage in case of any pregnancy/delivery complications?*'. The interview responses were marked against eight key steps required to achieve BPCR such as[1]: 'decided on the place of delivery—home or health facility'[2]; 'knew the facility that could provide emergency care'[3]; 'identified institution where to rush in case of emergency'[4]; 'identified people to accompany the woman',[5] 'identified people to take care of children at home'[6]; 'saved/arranged money for delivery expense or in case of emergency'[7]; 'advance arrangement of transportation to go to the facility' and[8] 'others—cloth, soap'.

#### Main exposure variable
The exposure variable, the IMFHL intervention, comprised four levels based on households' exposure to the IMFHL programme. An ordinal variable was created allowing us to evaluate the programme's main effect on the practice of BPCR, that is, the magnitude of change in BPCR practice across levels of IMFHL programme exposure: intervention (SHG plus health), comparison (SHG only) and pure control (no SHG, no health) households. The coding of the IMFHL explanatory variable, with a description of each group, is shown below:

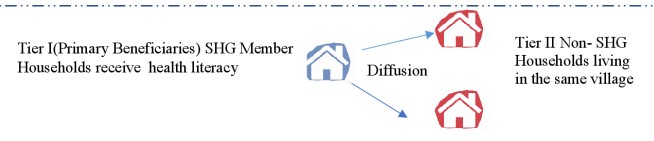

**Figure 1** Assumed pathway of knowledge diffusion from member households (tier I) to neighbouring non-member households (tier II) within villages. SHG, self-help group.

- ► Group 0: households who were not SHG members (non-members) and were in villages without *any* programme intervention (pure control households).
- ► Group 1: households who were not SHG members (non-members) but were in *programme villages* where either the SHG programme alone or SHG programme plus the health intervention was implemented (diffusion-control households).
- ► Group 2: households that were SHG members in programme villages where only the SHG programme was implemented (comparison households).
- ► Group 3: households who were SHG members and were in villages where both the SHG programme and the additional health intervention were provided. Only these households received health literacy intervention through the SHG (intervention households).

Additionally, a survey round variable was created to assess the effect of the intervention programme over time (change in women's BPCR in round II in 2017 compared with round I in 2015, when the programme was yet to be implemented).

### Confounding variables

A comprehensive set of confounding variables capturing individual, household and community-level characteristics was identified from previous maternal and child health literature and included in models III and IV. The confounding variables in model III represented eligible women's maternal health status and broader maternal health service utilisation indicators. Additional sociodemographic and area-level characteristics were included in the final model IV. Model IV also included a household wealth quintile variable, constructed for this analysis using polychoric principal component analysis, combining all household assets and amenities to evaluate BPCR association across five wealth gradients extending from marginally poor (reference category) to the poorest households. Covariates also included a variable to capture household poverty using the income threshold of 'Below Poverty Line (BPL)' card set by the Indian Government.

### Statistical analysis

A multivariable logistic regression modelling approach was used to evaluate the intervention programme's impact on the level of BPCR practice among women in rural UP, adjusting for various confounding variables. The main explanatory variable, IMFHL intervention, was categorised according to four levels of household exposure to the IMFHL programme, as briefly outlined below. Four separate regression models were fitted to the data, with the first model (model I) establishing the programme's main effect without any confounding variables but including the survey round variable. The second model (model II) included an interaction term (IMFHL intervention by survey round) to draw out the change over time in the effect of IMFHL programme exposure. Confounders related to individual health and health system were included in model III, while the full

model model IV included sociodemographic, economic and area-level variables. Thus, models I and II provide unadjusted programme effects while models III and IV provide adjusted effects. All analyses were performed using Stata V.16 (Statacorp, USA).

## RESULTS

### Descriptive statistics

Table 1 compiles the levels and steps of BPCR practice in the last pregnancy among women aged 15–49 in rural UP (n=17 244). As table 1 reveals, only 24% of all women in this analysis had not practised any steps for birth or complication readiness, while 49% have undertaken full BPCR in their last pregnancy. A further 6% of women had only taken steps to address birth preparedness, while 21% of all women had only undertaken the three steps required for complication readiness.

Table 2 presents the key descriptive statistics for eligible women categorised across SHG member and non-member households across all variables used in this paper. The study sample included more women from non-member households (59%) than from member households, but with an equal allocation of households within various levels of IMFHL programme exposure.

Sampled women had a mean parity of 2.4 (median 2, range 0–9), which reflects the fertility rate in rural UP, but it is higher than the current Indian fertility rate (median=2.2, range: 1–4).[38] Moreover, close to 48% of all surveyed women reported experiencing a pregnancy complication in their last pregnancy, while only a quarter (25%) had suffered a spontaneous or induced abortion. Furthermore, women on average had four contacts with frontline workers in their last pregnancy, with 26% of women receiving four ANC visits with all tests during their last pregnancy and only 9% of women receiving the vital three postnatal visits. The table compared SHG members and non-members using Chisqure and independent samples t test with statistical significance reported. While comparability is seen for key variables, all descriptive variables were adjusted in subsequent regression models (III–IV).

### Patient and public involvement

No patients or community members were directly involved in this study.

### Multivariable logistic regression analysis results: models I–IV

The results from multivariable logistic regression models are presented in table 3 for Models I and II (effects of IMFHL programme exposure and survey round), and in table 4 for models III and IV (effects of IMFHL programme exposure and survey round adjusting for sociodemographic, economic and area-level confounders). The OR, along with their associated 95% 95% CIs are reported. The a priori level of significance was set at the usual 5% alpha and all p values reported in tables using the asterisk convention: ***: p<0.01; **:

**Table 1** Levels of BPCR practice in last pregnancy among women aged 15–49 in rural Uttar Pradesh

| Level of preparedness | Number (%) (n=17 244) | Specific activity done |
|---|---|---|
| No BPCR preparedness | 4187 (24 %) | |
| Only birth preparedness done | 955 (6%) | 0. Decided on the place of delivery (at home or health facility) |
| | | 1. Knew the facility that could provide emergency care |
| | | 2. Identified people to accompany the woman |
| | | 3. Identified people to take care of children at home |
| | | 4. Others (oil/cloth/soap) |
| Only complication readiness done | 3697 (21%) | 1. Saved/arranged money for delivery expense or in case of emergency |
| | | 2. Advance arrangement of transportation to go to the facility |
| | | 3. Identified institution where to rush in case of emergency |
| Both birth preparation and complication readiness steps done | 8405 (49%) | 1. Decided on the place of delivery (at home or health facility) |
| | | 2. Knew the facility that could provide emergency care |
| | | 3. Identified people to accompany the woman |
| | | 4. Identified people to take care of children at home |
| | | 5. Saved/arranged money for delivery expense or in case of emergency |
| | | 6. Advance arrangement of transportation to go to the facility |
| | | 7. Identified institution where to rush in case of emergency |
| | | 8. Others (oil/cloth/soap) |
| Total | 17 244 (100%) | |

BPCR, birth preparedness and complication readiness.

p<0.05; *: p<0.10, with the last category meant to show that a 'trend towards statistical significance' has to be noted.

### Model I: IMFHL programme's effects on BPCR

Women belonging to an SHG only and those belonging to and receiving the health intervention from a SHG had higher odds of reporting birth and complication readiness practice in the last pregnancy than non-SHG members in non-programme (pure control) villages (table 3). Women in SHG households (members) were 1.17 times more likely to have practised BPCR in their last pregnancy (OR=1.17, 95% CI 1.07 to 1.29, p<0.01) than non-members in pure control villages. SHG members who received additional health literacy were 1.48 times more likely to practise BPCR (OR=1.48, 95% CI 1.35 to 1.63, p<0.01) than non-members in pure control villages. In contrast, results also showed that non-member women living in the same villages where either SHG or SHG plus health intervention was implemented were 0.89 times *less likely* to practise BPCR compared with non-member women in pure control villages, that is, women in those villages where no SHG or health intervention was implemented (OR=0.89, 95% CI 0.82 to 0.97, p<0.01). Overall, results showed that women in households interviewed in round II were almost 0.37 times *less likely* to have practised BPCR compared with households interviewed in round I, given all the levels of programme exposure (OR=0.37, 95% CI 0.35 to 0.39, p<0.01).

### Model II: interaction effect of IMFHL programme and survey round

Model II, table 3 shows that the inclusion of the interaction term rendered the women in SHG plus health households in round I with a statistically not significant 4% lower odds of practising BPCR compared with non-member women in control villages in round I (OR=0.96, 95% CI 0.84 to 1.11, p>0.10). As explained, the magnitude observed here corresponds to what is happening in round I (in model II), while what was observed in model I (1.48 times more likely to practise BPCR) corresponds to the magnitude of the exposure all rounds combined. And this is expected since there is no programme exposure in round I. The interaction of the variable survey round with household's level of microfinance exposure showed findings for each level of household IMFHL programme exposure in round 2 compared with the household's same exposure level in round 1.

The results from the two-way interaction in model II of round by households' microfinance exposure level showed that women in SHGs that received the health literacy intervention were 2.21 times more likely to practise BPCR in round II compared with the same household type in round I (OR=2.21, 95% CI 1.82 to 2.68, p<0.01). This means that women in the intervention group were roughly 2.122 (=0.96×2.21) (95% CI 1.52 to 2.97) times more likely to practise BPCR in round II compared with women in control villages. Similarly, the results showed that non-members in programme villages reported 1.73 times higher odds of BPCR practice in round II compared

**Table 2** Summary statistics of key variables by non-member households and SHG households

| Variable | | Summary statistics (N=17 244) | | |
|---|---|---|---|---|
| | | Non- member households | SHG house holds | Test of comparison |
| | *Independent variables: programme exposure characteristics* | | | |
| 1. | **Level of HH MF exposure** | 10 097 (59%) | 7147 (41%) | --- |
| | 0. HH in a village with **no SHG, no health** intervention (pure control—reference) | 3709 (37%) | --- | --- |
| | 1. **Non-member** HH in a village with **SHG only programme** | 3042 (30%) | --- | --- |
| | 2. **Non-member** HH in a village with **SHG plus health** intervention | 3346 (33%) | --- | --- |
| | 3. SHG member HH in a village with **SHG only intervention** | --- | 3623 (51%) | --- |
| | 4. **SHG plus health member** HH in a village with **SHG *plus* health** intervention | --- | 3524 (49%) | --- |
| 2. | **Evaluation survey round** | | | |
| | Round 1/baseline-2015 (=0) | 5454 (54%) | 3269 (45%) | *** |
| | Round 2/endline-2017 (=1) | 4643 (45%) | 3878 (54%) | |
| | *Independent variables: individual health and health system characteristics* | | | |
| 3. | Parity (number of previous pregnancies) of the EW | Mean=2.4 (SD=1.44) | Mean=2.4 (SD=1.41) | |
| 4. | Ew with any past history of pregnancy loss (due to spontaneous/induced abortion) | 2550 (25%) | 1890 (26%) | * |
| 5. | EW experienced any complication in last pregnancy/labour or post-partum | 4784 (47%) | 3437 (48%) | |
| 6. | EW's with correct knowledge of the minimum(four) number of ANC required during pregnancy | 3716 (37%) | 2761 (39%) | *** |
| 7. | EW received four or more ANC in last pregnancy with urine/blood pressure / weight/abdominal/ultrasound tested in last ANC | 2668 (26%) | 2077 (29%) | *** |
| 8. | EW reporting Institutional delivery | 8357 (83%) | 5948 (83%) | |
| 9. | **Duration (in hours) of stay in health facility immediately after delivery** | | | |
| | 1. Home delivery (reference) | 1740 (17%) | 1199 (17%) | *** |
| | 2. Discharged within 12 hours | 5603 (56%) | 4127 (58%) | |
| | 3. Discharged between 12 and 24 hours | 1134 (11%) | 795 (11%) | |
| | 4. Discharged between 24 and 48 hours | 720 (7%) | 506 (7%) | |
| | 5. Discharged between 48 and 72 hours | 283 (3%) | 146 (2%) | |
| | 6. Discharged after >72 hours | 617 (6%) | 374 (5%) | |
| 10. | EW who received three PNC in the first 7 days after delivery | 909 (9%) | 659 (9%) | |
| 11. | Number of contacts with ASHA/ANM/AWW/SHG in last pregnancy | Mean=4.0 (SD=5.4) | Mean=4.2 (SD=5.4) | |
| 12. | Distance (kilometres) to primary health centre if not available in the village | Mean=5.4 (SD=4.9) | Mean=5.4 (SD=4.7) | |
| | *Independent variables: socio-demographic/economic and area level characteristics* | | | |
| 13. | Village distance (kilometres) to closest town | Mean=1.4 (SD=0.77) | Mean=1.4 (SD=0.75) | |
| 14. | Population of village | Mean=5153 (SD=5134) | Mean=5140 (SD=5113) | |
| 15. | HH with BP) card | 4499 (45%) | 3316 (46%) | |
| 16. | **Household wealth quintile (poor to poorest)** | | | |
| | 1. Marginally poor | 1985 (20%) | 1498 (21%) | *** |
| | 2. Moderately poor | 2049 (20%) | 1502 (21%) | |
| | 3. Poor | 1994 (20%) | 1471 (20%) | |
| | 4. Poorer | 2029 (20%) | 1396 (20%) | |
| | 5. Poorest | 2040 (20%) | 1280 (18%) | |
| 17. | EW presently working to earn cash, in kind or both | 1661 (16%) | 1211 (17%) | |

Continued

**Table 2** Continued

| | | Summary statistics (N=17 244) | | |
| | | Non- member households | SHG house holds | Test of comparison |
|---|---|---|---|---|
| 18. | EW living with joint and extended family | 5819 (58%) | 4259 (60%) | *** |
| 19. | Household head's regligion: Hinduism and others | 9281 (92%) | 6546 (92%) | |
| 20. | EW belonging to scheduled caste and scheduled tribe | 4485 (45%) | 3202 (45%) | |
| 21. | EW age in completed years | Mean=25 (SD=4.56) | Mean=25 (SD=4.55) | |
| 22. | EW's education level: completed primary/middle school (up to year 9) and above | 6680 (66%) | 4798 (67%) | |
| 23. | EW's husband education: completed primary/middle school (up to year 9) and above | 8365 (83%) | 5936 (83%) | |
| | *Dependent variable* | | | |
| 24. | EW's who practised BPCR in last pregnancy | 4662 (46%) | 3743 (52%) | *** |

ASHA/ANM and AWW are government health workers in villages as per population guidelines providing preventative maternal, child and other health services. Independent sample T-test and Chi-square test for group (SHG vs non-SHG) comparison with significant p-value shown as ***: $p<0.01$, **: $p<0.05$,* and $p<0.10$.
Non-member households are those which do not include a SHG member and SHG households are those that include a SHG member.
ANC, antenatal check-up; ANM, auxiliary nurse midwife; ASHA, accredited social health worker; AWW, anganwadi worker; BPCR, birth preparedness and complication readiness; BPL, below poverty line; EW, eligible woman; EW, eligible woman; HH, household; MF, microfinance; PNC, postnatal care visit; SHG, self-help group.

with the same household type in round I (OR=1.73, 95% CI 1.46 to 2.05, p<0.01). These results indicate that a diffusion effect occurred from SHG women receiving health literacy to non-member women in the same villages, with women in this group being almost 1.194 (=0.69×1.73) (95% CI 0.89 to 1.57) times more likely to practise BPCR.

**Table 3** Logistic regression models I and II results estimating levels of BPCR: ORs and associated 95% CI

| | | Model I | Model II |
|---|---|---|---|
| | **Explanatory variable name** | OR (95 % CI) | OR (95 % CI) |
| | **Main effects** | | |
| 1. | **Level of HH MF exposure** | | |
| | 0. HH in a village with **no programme** intervention (pure control) | Reference | Reference |
| | 1. **Non-member** HH in village with **SHG or SHG plus health intervention** | 0.89*** (0.82 to 0.97) | 0.69*** (0.61 to 0.77) |
| | 2. **SHG member** HH in village with **SHG only intervention** | 1.17*** (1.07 to 1.29) | 1.13* (0.98 to 1.30) |
| | 3. **SHG plus health member** HH in village with **SHG *plus* health** intervention | 1.48*** (1.35 to 1.63) | 0.96 (0.84 to 1.11) |
| 2. | **Round** | | |
| | Round 1 | Reference | Reference |
| | Round 2 | 0.37*** (0.35 to 0.39) | 0.25*** (0.22 to 0.29) |
| 3. | **Two-way interaction effects: Round # HH MF exposure** | **Interaction term effects** | |
| | Round 1 Non-MF HH in pure control village | | Reference |
| | Round 2 Non-MF HH in a village with MF or MF plus health | ---- | 1.73*** (1.46 to 2.05) |
| | Round 2 **SHG member** HH in village with MF programme only | ---- | 1.10 (0.90 to 1.33) |
| | Round 2 **SHG plus health member** HH in village with MF plus health intervention | ---- | 2.21*** (1.82 to 2.68) |

CI in parentheses; significant p-value shown as ***: p<0.01, **: p<0.05 and *: p<0.10.
BPCR, birth preparedness and complication readiness; HH, household; MF, microfinance; SHG, self-help group.

**Table 4** Logistic regression models III and IV results estimating levels of BPCR using confounders: ORs and associated 95% CI

| Serial number | Explanatory variable name | Model III<br>OR<br>(95% CI) | Model IV<br>OR<br>(95% CI) |
|---|---|---|---|
| 1. | Level of HH MF exposure | | |
| | 1. HH in a village with no programme intervention (pure control) | Reference | Reference |
| | 2. **Non-member** HH in a village with **SHG or SHG plus health intervention** | 0.69*** (0.61 to 0.78) | 0.70*** (0.62 to 0.78) |
| | 3. **SHG member** HH in a village with **SHG only intervention** | 1.13* (0.98 to 1.30) | 1.14* (0.99 to 1.31) |
| | 4. **SHG plus health member** HH in a village with **SHG plus** health intervention | 0.97 (0.84 to 1.11) | 0.98 (0.85 to 1.12) |
| 2 | **Round:**(Round 1) | Reference | Reference |
| | Round 2 | 0.24*** (0.21 to 0.28) | 0.24*** (0.20 to 0.27) |
| 3. | **Two-way interaction effects (round # SHG exposure)**<br>*(Round 1# Non-MF HH in pure control village)* | Reference | Reference |
| | Round 2 # Non-MF HH in village with MF or MF plus health | 1.72*** (1.45 to 2.04) | 1.72*** (1.45 to 2.04) |
| | Round 2 MF-HH in village with MF only | 1.10 (0.90 to 1.33) | 1.10 (0.90 to 1.33) |
| | Round 2 MF plus health HH in village with MF plus health | 2.20*** (1.82 to 2.67) | 2.20*** (1.81 to 2.66) |
| | *Model III using individual health and health system confounders* | | |
| 4. | **Parity (number of previous pregnancies) of EW** | 0.98 (0.96 to 1.00) | 0.98 (0.95 to 1.01) |
| 5. | **Any past pregnancy loss (due to spontaneous/induced abortion)** : (no previous pregnancy loss) | Reference | Reference |
| | Previous pregnancy loss | 0.98 (0.91 to 1.05) | 0.98 (0.90 to 1.05) |
| 6. | **Any complication experienced in last pregnancy/labour or postpartum:**(no complication experienced) | Reference | Reference |
| | Complication experienced | 0.95 (0.89 to 1.01) | 0.95 (0.89 to 1.01) |
| 7. | **EW's knowledge of the minimum number of ANC required during pregnancy:** (incorrect knowledge) | Reference | Reference |
| | Correct knowledge | 1.07** (1.00 to 1.14) | 1.06* (0.99 to 1.13) |
| 8. | **Received four or more ANC in last pregnancy (with urine/blood pressure / weight/abdominal/ultrasound tested in last ANC):** (not received) | Reference | Reference |
| | Received four ANC with all tests done in last ANC | 0.92 (0.85 to 1.01) | 0.91** (0.82 to 0.99) |
| 9. | **Place of last delivery:** (home delivery) | Reference | Reference |
| | Institutional delivery | 0.95 (0.81 to 1.11) | 0.92 (0.79 to 1.08) |
| 10. | **Duration (in hours) of stay in health facility immediately after delivery:**(home delivery) | Reference | Reference |
| | Discharged within 12 hours | 1.14* (0.98 to 1.31) | 1.15** (0.99 to 1.33) |
| | Discharged between 12 and 24 hours | 1.09 (0.92 to 1.29) | 1.10 (0.93 to 1.30) |
| | Discharged between 24 and 48 hours | 0.98 (0.82 to 1.17) | 0.99 (0.83 to 1.18) |
| | Discharged between 48 and 72 hours | 1.02 (0.81 to 1.29) | 1.05 (0.83 to 1.32) |
| | Discharged after >72 hours | 1.00 | 1.00 |
| 11. | **Received three PNC in first 7 days after delivery:**<br>(Not received any PNC or received after 7 days) | Reference | Reference |
| | Received three PNC in first 7 days after delivery | 1.06 (0.95 to 1.19) | 1.05 (0.94 to 1.18) |
| 12. | **Number of contacts with ASHA/ANM/AWW/SHG in last pregnancy** | 0.99 (0.99 to 1.00) | 1.00 (0.99 to 1.00) |
| 13. | **Distance to primary health centre if not available in the village** | 0.99 (0.98 to 1.00) | 0.99 (0.99 to 1.00) |
| | *Model IV using socio-demographic/economic and area-level confounders* | | |
| 14. | **Village distance (kilometres) to closest town** | ---- | 0.99 (1.03 to 1.13) |
| 15. | **Population of village** | ---- | 1.00 (0.99 to 1.00) |
| 16. | **HH with BPL card:** (no—HH does not have BPL card) | ---- | Reference |
| | Yes—HH has BPL card | ---- | 1.01 (0.94 to 1.08) |
| 17. | **Household wealth quintile (poor to poorest)** | | |
| | 1. Marginally poor | ---- | Reference |
| | 2. Moderately poor | ---- | 1.11** (1.00 to 1.22) |

**Table 4** Continued

| Serial number | Explanatory variable name | Model III OR (95 % CI) | Model IV OR (95 % CI) |
|---|---|---|---|
| | 3. Poor | ---- | 1.00 (0.90 to 1.10) |
| | 4. Poorer | ---- | 0.91* (0.81 to 1.01) |
| | 5. Poorest | ---- | 0.87** (0.77 to 0.97) |
| 18. | **EW presently working to earn cash, in-kind or both:** (not working) | ---- | Reference |
| | Currently working | ---- | 1.00 (0.92 to 1.09) |
| 19. | **Family type:** (nuclear) | ---- | Reference |
| | Joint and extended | ---- | 1.05 (0.98 to 1.13) |
| 20. | **Religion:** (Muslim) | ---- | Reference |
| | Hinduism and others | ---- | 1.06 (0.94 to 1.19) |
| 21. | **Scheduled caste:** (general caste) | ---- | Reference |
| | Other backward castes | ---- | 0.96 (0.87 to 1.07) |
| | Scheduled caste and scheduled tribe | ---- | 0.97 (0.87 to 1.08) |
| 22. | **EW age in completed years** | ---- | 1.00 (0.99 to 1.01) |
| 23. | **EW education level:** (no schooling) | ---- | Reference |
| | Completed primary/middle school (up to year 9) and above | ---- | 0.96 (0.89 to 1.03) |
| 24. | **EW's husband education level:** (no schooling) | ---- | Reference |
| | Completed primary school/middle school (up to year 9) and above | ---- | 1.06 (0.97 to 1.17) |
| | **Estimation of model fit** | **Model III** | **Model IV** |
| | Log likelihood | −11 354 | −11 334 |
| | Number of observations | 17 244 | 17 244 |
| | AIC/BIC | 22 750/22 914 | 22,741/23 021 |

CI in parentheses; significant p-value shown as ***: p<0.01, **: p<0.05 and *: p<0.10. Log-likelihood and AIC/BIC values are also reported.
AIC (Akaike's Information Criteria) and BIC (Bayesian Information Criteria) are used as model selection criteria.
ANC, antenatal check-up; BPCR, birth preparedness and complication readiness; BPL, below poverty line; EW, eligible woman; HH, household; MF, microfinance; PNC, postnatal care visit; SHG, self-help group.

## Models III and IV: effects of IMFHL programme adjusted for confounders

Table 4 shows model III and IV results where the effect of main exposure variable, IMFHL programme exposure and survey round, and interaction term is assessed when confounders are added. Overall, the effects of programme exposure on BPCR observed previously in model II remained unaffected when adjusting for confounders in models III and IV, suggesting that the outcome, BPCR, is mainly influenced by programme exposure rather than confounders.

### Key confounders: effects of individual health and health system variables

Women who were discharged earlier from the health facility after delivery had higher odds of reporting the practice of all steps of BPCR in their last pregnancy (OR=1.15, 95 % CI 1.00 to 1.33, p<0.05). In contrast, women who reported receiving four ANC visits with key tests done in the last ANC visit reported *lower odds* of BPCR practice compared with women who received less than four visits and with incomplete tests (OR=0.90, 95 % CI 0.82 to 0.99, p<0.05). Among variables reflecting the continuum of maternal care, women with the correct knowledge about the required number of ANC visits were more likely to practise BPCR compared with those with incorrect knowledge (OR=1.06, 95 % CI 0.99 to 1.13, p<0.10).

### Key confounders: effect of sociodemographic, economic and area-level variables

Model IV results for the wealth quintile variable showed that the likelihood of BPCR practice reduced as the level of household poverty increased from marginally poor to the poorest households, with the poorest households (those in the fifth wealth quintile) being 0.87 times less likely to practise BPCR than women in marginally poor households (those in the first quintile). Maternal education, which has previously been found in the literature to be strongly correlated with routine maternal health behaviours, in contrast, was negatively associated with BPCR practice in this analysis. Mothers who had completed at least primary education were 7% less likely to practise BPCR than mothers without any schooling (OR=0.93, 95 % CI 0.87 to 0.99, p<0.05).

## DISCUSSION

This study makes a significant contribution to the literature on maternal health promotion and implementation by investigating the main effect of IMFHL programme

membership on BPCR practice among poor women who have recently given birth in rural UP. The study provides evidence of the impacts of SHG membership alone and when integrated with health literacy on BPCR practice. This evidence suggests that BPCR practice is potentially improved during maternal complications among SHG members when SHGs were integrated with the health literacy programme and supports the claim that microfinance-based women's groups are an effective strategy to provide health literacy to marginalised rural populations enabling behaviour change. No previous study has examined the impact on BPCR in the context of SHG programmes alone or in an integrated SHG and health literacy intervention programme in India or elsewhere. This is also the first study to find the presence of a diffusion effect of maternal health promotion behaviour—BPCR practice—from SHG members receiving health literacy to non-members in programme villages.

Overall, the results reveal that SHGs promote the adoption of BPCR practice among members and the odds of BPCR adoption almost doubles when a health literacy component is integrated into the model. However, the diffusion effect of BPCR practice is only observed when SHGs have an integrated health literacy component. The study finds the diffusion of BPCR practices among non-members in SHG plus health literacy villages and not in the SHG-only villages.

Therefore, these findings suggest that SHGs require the addition of a maternal health literacy component to achieve a higher coverage of a behavioural health intervention among members and non-members in villages. The underlying mechanism of diffusion requires additional research to explain pathways; however, insights from previous studies suggest that either SHG members may themselves be promoting practices as 'model adopters' of new behaviour through interpersonal contact, or that SHGs serve as a better platform for facilitating linkages between non-members and community health workers when combined with health literacy intervention.[36 37 39] It is, therefore, feasible that when SHG members receive health literacy, they may be acting as change agents to encourage non-members to adopt desired health behaviours. The study further highlighted that complex health behaviour such as BPCR, which comprises multiple steps to constitute full practice, in the absence of programme intervention, declined with time reflecting an overall negative secular effect of time. The IMFHL programme, thus, has a protective effect of pulling up these lagging indicators, which are otherwise likely to keep decreasing and negatively impact maternal health.

This study highlights the important role that SHGs play as a catalyst for promoting maternal health behavioural change when layered up with health literacy intervention. The findings are supported by other studies from UP and Bihar, another Indian state with low development indicators, which had shown an increase in routine indicators of maternal health system utilisation (such as four ANC visits and institutional delivery) and home-based neonatal healthcare

practices by 5 to 11 percentage points when health literacy intervention was provided through SHGs compared with membership in SHGs alone.[33 40] In this study, the higher likelihood of BPCR practise among women in SHGs plus health intervention households could be a direct consequence of SHG membership that provides a platform, which gives women access to needed funds. This access addresses one of the key BPCR steps—saving funds for a maternal emergency that influences care-seeking behaviours in low-income families. High maternal health expenses are seen as a strong factor for pushing vulnerable families at risk of poverty into debt traps that reinforce debt, borrowing and illness cycles. Therefore, in such settings, the provision of health literacy and financial access together are expected to improve households' care-seeking preferences. Previous studies have also shown that SHGs provide mutual support among members.[33 36] Consequently, health literacy delivered through the peer network of SHGs is likely to change health practices among marginalised women.

In adjusting for other socioeconomic confounders, this study also found that across wealth quintiles, a significant negative effect in BPCR practices was observed only in the poorest (fifth quintile) households, with these households least likely to practise BPCR.

## Study strengths and limitations

This was a first-of-its-kind study in rural India that comprehensively evaluated the impact of an IMFHL programme on uptake of BPCR practice, which can reduce delays in maternal care-seeking during complications.

A key strength of the study relates to the large sample size of 17 244 women representing marginalised rural populations compared with other studies. Additionally, the study adds value to community evaluation research by using a quasi-experimental study design with preintervention and postintervention measurements, which is potentially the efficient approach in a highly populated developing country setting, providing the opportunity to evaluate the incremental change across programme exposure levels (impact of SHG only, SHG plus health intervention) and the potential diffusion effect (impact of SHG plus health intervention from members to non-members in areas without any programme exposure).

Investigating the diffusion effect of practice from groups who received the intervention to neighbouring non-member households generated evidence on the expanded reach of such an intervention, leading to greater coverage of vulnerable populations. The key findings, however, should be interpreted while considering the following study limitations.

First, the selection of programme blocks under the IMFHL was based on the programme's operational criterion, thus preventing randomisation. The quasi-experimental survey design, however, adopted a multistage sampling approach using different criteria to select blocks (Scheduled Caste/Scheduled Tribe [SC/ST] percentage) and villages (SHG coverage and population size) to limit selection bias and

allow for comparisons of areas with similar sociocultural and economic characteristics.

Also, the survey collected information from women based on self-recall, which may suffer from recall bias and social desirability. However, the potential for these errors is minimised as the women were interviewed within 12 months post-delivery and by trained interviewers.

The IMFHL programme's definition of SHG membership, which consisted of the pregnant woman herself or her mother-in-law or sister in law is an important consideration for understanding how the health literacy component informed the pregnant woman's health behaviour change in intervention areas. In intervention areas, the pregnant woman received health information directly from the SHG whether she was a member or lived in a household where the mother in law or sister in law were members. The pregnant woman was invited to attend health literacy meetings in the SHG. In the programme, the majority(two-thirds) of the pregnant woman was SHG members themselves, and (two-thirds) lived in a household where the mother in law or sister in law was the member. Thus, for some pregnant women, the SHG platform may have a broader role in involving other family members in the health literacy discussions. This is an important consideration in rural UP where family members, especially the mother-in-law, influence the adoption of maternal health behaviours.

Finally, while the survey design selected women from SHG member and non-member households from the same villages, in some cases, the women sampled in a given village would be either all members or all non-members, which potentially limited the variation in the outcome variable within villages.

## Policy and programme implications

A programmatic strength of the IMFHL intervention was that it used the existing structure of the SHG programme to embed the health literacy component, which provided evidence of a scalable model for disseminating health information. The findings reported in this study are relevant for the rural Indian context, where the Indian Government is establishing a network of women-only SHGs as a national poverty alleviation programme. Therefore, allowing the integration of the health intervention into SHG structures, as the IMFHL programme creates the opportunity to improve maternal health outcomes for the state and the country.

Finally, as healthcare-seeking behaviours are shaped by cultural practices in community settings, the diffusion effect of health messages from members to non-members can reinforce desired health behaviours. Diffusion also increases programme coverage of harder to reach populations. Future research is required to explore the pathways of the diffusion effect of knowledge and practice from members to non-members in programme villages.

## CONCLUSIONS

The IMFHL programme evaluated in this study promoted behaviour changes among some of the most marginalised households in UP, India's most populous state and one of its most developmentally disadvantaged areas. The findings show that SHGs exert both a dissemination effect of planned health behaviour within members as well as a diffusion effect of the natural transfer of BPCR practice from members to non-members but only when a health literacy component is added to SHGs. The findings from this research support integrating health literacy components into SHG networks and similar community groups to promote households' adoption of complication readiness plans to reduce delays in seeking healthcare during maternal complications.

Lessons from the IMFHL programme implementation in India provide the opportunity to adapt SHG models implemented elsewhere by adding health literacy components. Beyond India, microfinance is promoted globally as a poverty alleviation programme in low-resource settings where the burden of maternal mortality remains high. These types of interventions' can assist other low-resource settings in accelerating progress towards the maternal health target of the SDGs.

**Acknowledgements** We wish to thank Professor Rachel Davey, Health Research Institute, the University of Canberra, for her guidance and for reviewing the final draft of this manuscript. The authors would also like to thank the University of Canberra (Australia), the Public Health Foundation of India (India),the Indian Institute of Public Health-Gandhinagar(India) and the Population Council (India) for supporting this international research collaboration.

**Contributors** DA conceptualised the study, conducted the analysis, interpreted the results and drafted the study. DA acts as the paper's guarantor. IM guided the study design, the data analysis, the interpretation of the results and discussion. TN guided the study design, the data analysis and interpretation of results and discussion. All authors read and approved the final manuscript.

**Funding** The paper was part of the PhD research of the first author (Danish Ahmad), who was supported by the University of Canberra's Higher Degree by Research scholarship for his PhD. The Bill and Melinda Gates Foundation provided open access funding for the publication.The IMFHL program was supported, in whole, by the Bill & Melinda Gates Foundation [INV-006915].

**Competing interests** None declared.

**Patient and public involvement** Patients and/or the public were not involved in the design, or conduct, or reporting, or dissemination plans of this research.

**Patient consent for publication** Not applicable.

**Ethics approval** This study involves human participants and was approved by Ethics approval was granted by the Human Research Ethics Committee, at the University of Canberra for the program '2266 - PhD:The Impact of An Integrated Health Literacy and Microfinance Program on Care Awareness and Care Seeking Behaviour During Maternal Complication: The Case of Rural India'. Previously, an independent ethics committee of Population Council, New York, USA, approved the primary data collection under the IMFHL program. Participants gave informed consent to participate in the study before taking part.

**Provenance and peer review** Not commissioned; externally peer reviewed.

**Data availability statement** Data are available upon reasonable request. The datasets used and/or analysed during the current study are available from the corresponding author on reasonable request.

**ORCID iDs**
Danish Ahmad http://orcid.org/0000-0001-7891-3756

Theophile Niyonsenga http://orcid.org/0000-0002-6723-0316

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
