## [Reviewer comments · BMJ Open]

ARTICLE DETAILS

TITLE (PROVISIONAL)	Improving birth preparedness and complication readiness in rural India through an integrated microfinance and health literacy program: Evidence from a quasi-experimental study
AUTHORS	Ahmad, Danish; Mohanty, Itismita; Niyonsenga, Theophile

VERSION 1 – REVIEW

REVIEWER	Saaka, Mahama University for Development Studies, Community Nutrition
REVIEW RETURNED	25-Jul-2021

GENERAL COMMENTS	Comments on Manuscript Entitled “Improving birth preparedness and complication readiness in rural India through an integrated microfinance and health literacy program: Evidence from a quasi-experimental study” GENERAL COMMENTS The study sought to reduce the persistent high levels of maternal mortality rate (MMR) through the promotion of birth preparedness and complication readiness (BPCR) among pregnant women. The study specifically evaluated the impact of membership in an integrated microfinance and health literacy (IMFHL) program on the practice of BPCR in rural India. The following are some concerns that need to be addressed to help improve the manuscript. ABSTRACT The abstract is well written with all the essential elements included. The primary was BPCR but it is not clear whether what the “secondary outcome” was. If there is no secondary outcome, then the words should be deleted. INTRODUCTION The introduction was well written, and the problem statement articulated to indicate knowledge gap the study sought to address. However, it could have been shortened to about one and a half pages. The focus of the paper was on how to improve birth preparedness and complication readiness (BPCR) as strategy to reduce maternal mortality. So, the first paragraph can be summarized to about 3-4 sentences that will highlight the magnitude of and consequences of maternal mortality. METHODS Study aim and design i. The aim and objectives of the study should not be in the methodology section. Usually and preferably, they should be stated in the last paragraph of the introduction section. The authors should therefore remove the study objectives from this section.ii. If the IMFHL program was implemented over five years (2012–2017), why is it stated on page 7 that “ data was collected as part of the IMFHL project in two rounds (pre-post program implementation), capturing the program’s characteristics and other factors in the first round in 2015”. Why 2015 and not 2012? Revise to make clear.
---

iii. Additionally, a lot of the stuff written under this section, are not study design matters. For example, the stuff starting from “The maternal and neonatal health literacy component was layered upon a microfinance-based SHG platform towards 2015 to provide pregnant and recently delivered women with health messages in rural UP. A detailed description of the IMFHL program context, selection of intervention, comparison and control blocks is available elsewhere (38,39)..... This diffusion of knowledge from members (tier I) to neighbouring non-members (tier II) is depicted in Figure 1 below”.

iv. The authors are advised to remove this information and place it under a different sub-heading as “Description of the Intervention”

Sample size and sampling procedure

It is stated on page 9 that a detailed description of the selection strategy is available elsewhere (38,39). A brief description of the sampling procedure for households is still needed in this paper. Similarly, though details of the sample size determination are provided elsewhere, the reader of this paper needs to be informed of how this was done especially when he/she has no access to that information.

Statistical analysis

For the sake of order and logical flow of study activities, the statistical analysis section should come after the section on measurement of outcome, exposure, and confounding variables. What statistical package was used for data analysis? SPSS? Stata? etc.

RESULTS

The “Descriptive Statistics” should follow a major heading titled “RESULTS”

i. It is also important to compare the socio-demographic characteristics of study participants at baseline before the substantive results of the study. Any baseline differences not accounted for will confound the outcome of the study.

ii. Table 2 is too crowded and confusing because different information types including sociodemographic characteristics, antenatal attendance, individual health and health system characteristics of the study participants are all contained in the table. The table should be split appropriately into two distinct tables under separate sub-headings as follows: sociodemographic characteristics of the sample, past obstetric data and medical history of mothers (e.g., antenatal attendance, place of delivery etc.). It will be informative to add a measure of association (e.g., chi-squared statistic) between these variables and type of household (non-member households and SHG) presented in Table 2.

iii. I am wondering whether the program exposure characteristics should be presented in Table 2 since they are not comparable across the different households.

iv. The information presented in Tables 1 and 2 are also part of study results and I am therefore wondering why they were presented before the section titled “RESULTS”. Is it supposed to suggest that only the information presented in tables 3-4 are the results of the study?

v. How was additive effect of health literacy on BPCR isolated from that SHG households membership alone? For example, women in SHG households were 1.17 times more likely to have practised BPCR in their last pregnancy (OR=1.17, 95% CI: 1.07–1.29, $p<0.01$) than non-members in pure control villages. SHG members who received additional health literacy were 1.48 times more likely to practise BPCR (OR=1.48, 95% CI: 1.35–1.63, $p<0.01$) than non-members in pure control villages. These were generated from two separate regression models. What is not clear is what will happen when all models are combined into one? What will be the independent effect of all four exposures in regression model?

	vi. Interestingly, non-member women living in the same villages where either SHG or SHG plus health intervention was implemented were 0.89 times less likely to practise BPCR compared to non-member women in pure control villages (OR=0.89, 95% CI:0.82–0.97, p<0.01). vii. It is also reported that overall, results showed that women in households interviewed in round II were almost 0.37 times less likely to have practised BPCR compared to households interviewed in round I given all the levels of program exposure (OR=0.37, 95% CI:0.35–0.39, p<0.01). Were these women in the programme or non-programme communities? I guess this apparent negative effect arose because different households were involved in the baseline and follow-up surveys. I think this is a major limitation of this intervention study. viii. I have reservations for the interpretation of some of the results. For example, on page 18 it was stated “the results showed that non-members in program villages reported 1.73 times higher odds of BPCR practice in round II compared to the same household type in round I (OR=1.73, 95%CI: 1.46–2.05, p<0.01)”. These results were obtained from different groups of women at different times (baseline and follow-up surveys). Since it was not the same group of women that were followed to round II, it will be difficult to conclude that a diffusion effect occurred from SHG women receiving health literacy to non-member women in the same villages, with women in this group being almost 1.194 (=0.69*1.73) (95%CI:0.89-1.57) times more likely to practise BPCR. I am of the conviction that the observed effect could be due merely to secular trends or just group differences and not to programme effect. A better evidence could be obtained if the comparison was made with different women in different non-programme communities. It should be remembered that the authors have earlier stated that non-member women living in the same villages where either SHG or SHG plus health intervention was implemented were 0.89 times less likely to practise BPCR compared to non-member women in pure control villages (OR=0.89, 95% CI:0.82–0.97, p<0.01). So, is it right to attribute any diffusion effect to the intervention? ix. Are the odds ratios presented in Table 3 crude or adjusted? x. In Table 4, there are insignificant predictors of BPCR in the regression models. Usually, insignificant variables are removed from the analysis since their presence does not give the true measures of association of the independent variables. The authors should re-run the regression and remove the non-significant variables in order to have a better understanding of which factors really were independent predictors of BPCR. xi. Four separate regression models were fitted to the data, The second model (Model II) included an interaction term (IMFHL intervention by survey round) to draw out the change over time in the effect of IMFHL program exposure. What is not certain is whether in Model III and the full model, Model IV the interaction term was included in the modelling since that is not clear from table 4. xii. The coefficient of the interaction term (time x treatment) will give the effect of the interventions with time. The full model is thus expected to contain main independent variable which in this case is the IMFHL intervention, all the independent covariates and the interaction term. The presentation of the results needs to be improved so as to understand the effect of the intervention over time. I suggest a table like the following will provide clarity to the results.
--	---

REVIEWER	Mehta, KM University of California San Francisco
REVIEW RETURNED	31-Aug-2021

GENERAL COMMENTS

This article is about a very important topic, that is, the combination of microfinance and health layering to achieve maternal and child health outcomes. This paper focuses on a particular outcome, namely uptake of birth preparedness at the time of delivery. Some clarifications to the presentation of the analyses performed and the interpretation of the results are warranted. Below please find my detailed comments:

1. How does the cross sectional analysis align with the IMF HL program timing? See below comments, this seems like 2 serial cross sectional surveys?
2. Is this a healthy layering upon existing microfinance intervention?
3. Do you have self help groups with health layering and self-help groups without was a statistical comparison made between these two?
4. How was the diffusion effect evaluated? What is the underlying theory of change?
5. Though this trial is non randomized can you delete me 8 what are the main sociodemographic differences?
6. What is comprised in the birth preparedness and complication readiness intervention ?
7. Once the health layering and the financed based assistance given to self-help group women at the same time?
8. P 9 of 40, top, it is unclear why 2015 data was used if data were collected from 2015-2017
9. Page 9 of 40 lines 15 through 20 this is a problematic definition of recipient, if I'm mother-in-law or sister-in-law is given the information then no direct information is given to the one woman who needs the knowledge please state this as a limitation
10. page 9 of 40 lines 31 through 35 are these women incentivized to diffusing content or knowledge in their network
11. Page 11 of 40 lines one through 8 the statistical method described here including an interaction term to draw out the change overtime would suggest that the study design is a serial cross sectional study not a single cross sectional study
12. Page 11 of 14 lines 19 through 28 the outcome variable is reported for the last pregnancy what is a bit unclear is weather this last pregnancy would have occurred after the I am FHL intervention do you have specific birth date of the children to verify that the intervention took place prior to the birth?
13. Page 11 of 14 lines 38 through 49 did the BPCR training cover all key eight steps listed here was there any vet verification of knowledge acquisition?
14. Page 13 of 14, lines 6-11, how were these different program exposures captured? By geography?
15. Page 13 of 15 lines 43 through 47 it is unclear how models three and four are parameterized please give more details
16. page 18 line 22 the respondents educational level at 66% completing primary middle school up to year nine seems very high do you mean this is any educational level up to 9th standard?
17. Page 18 line 24 did you consider analyzing no BPCR steps as the reference then having a category of incomplete BPCR and lastly, comparing completed BPCR steps practiced in pregnancy
18. Page 19 line 38 the finding that nonmember women living in shg villages for .89 times as likely to practice BPCR compared to non member women in control villages this means that non member met women in control villages actually had better BPCR why do the authors think this is the case were there any Co interventions that may have influenced BPCR in these communities?
19. Page 19 blind 44 through 49 one reason that you may have found less practiced BPCR at follow-up or in round two compared to round one it is because of attrition it may be that those individuals who practiced BPCR or differentially last to follow up please state this as a consideration in the discussion

	20. Page 19 end face odds ratio 0.96 is not 4% lower odds as the confidence bounds include one this odds can be interpreted as statistically similar 21. page 20 top lines four through 11 I find this description confusing please clarify 22. Page 20 middle paragraph I suggest creating it table that has data based on a similar analysis in round one and round two this will clarify what model you are comparing in this paragraph 23. Page 20 lines 32 through 37 again the odds ratio did picked it here 1.194 with an interval that includes one can be interpreted as statistically similar 24. Page 20 2 model 3 I think the strongest finding in the paper is for round two SHG plus health member and this finding should be highlighted in the abstract 25. Page 26 top I would be careful or cautious about the interpretation of the diffusion there may be many reasons that non SHG members in SHG plus health literacy villages have an increased effect perhaps it would be good for the authors to explore some examples I am confident that if these questions are addressed fully, this paper would make a very nice contribution to the literature.
--	--

VERSION 1 – AUTHOR RESPONSE

Reviewer #1(R1): Dr. Mahama Saaka, University for Development Studies

The study sought to reduce the persistent high levels of maternal mortality rate (MMR) through the promotion of birth preparedness and complication readiness (BPCR) among pregnant women. The study specifically evaluated the impact of membership in an integrated microfinance and health literacy (IMFHL) program on the practice of BPCR in rural India. The following are some concerns that need to be addressed to help improve the manuscript

Response: We thank the reviewer for this comment.

ABSTRACT

R1-1 Reviewer Comment . *The abstract is well written with all the essential elements included. The primary was BPCR but it is not clear whether what the “secondary outcome” was. If there is no secondary outcome, then the words should be deleted.*

Response: Thank you for highlighting this Dr Saaka; we have removed the term ‘secondary outcomes’ in the abstract on page 2, line 36, which was initially kept as per journal formatting suggestion.

INTRODUCTION

R1-2 Reviewer Comment : The introduction was well written, and the problem statement articulated to indicate knowledge gap the study sought to address. However, it could have been shortened to about one and a half pages. The focus of the paper was on how to improve birth preparedness and complication readiness (BPCR) as strategy to reduce maternal mortality. So, the first paragraph can be summarised to about 3-4 sentences that will highlight the magnitude of and consequences of maternal mortality.

Response: Authors thank Dr Saaka for the suggestion. We believe addressing this query in the revised draft will improve the overall presentation of the paper. We have accordingly reduced the introduction to close to 2 pages while retaining necessary background information. The paper presents multidisciplinary research spanning maternal health, community program development and microfinance and thus seeks to provide adequate contextual information in the introduction to suit a potentially wide readership in academic and public health program implementation.

METHODS

Study aim and design

R1-3: Reviewer Comment : The aim and objectives of the study should not be in the methodology section. Usually and preferably, they should be stated in the last paragraph of the introduction section. The authors should therefore remove the study objectives from this section.

Response: Authors thank the reviewer for this comment. We have now addressed this issue in the revised draft with study aims now stated in the last paragraph of the introduction section on page 7 lines 160-166

R1-4: Reviewer Comment : If the IMFHL program was implemented over five years (2012– 2017), why is it stated on page 7 that" data was collected as part of the IMFHL project in two rounds (pre-post program implementation), capturing the program's characteristics and other factors in the first round in 2015". Why 2015 and not 2012? Revise to make clear.

Response: We have noted the query; we are sharing a clarification and description of changes made. The IMFHL program was planned across two phases with a pilot phase from 2012 to 2015 in limited areas of UP state, during which the final program implementation strategy was revised. The final program implementation was conducted at scale from 2015 to 2017 in different geographies of UP state. We have added these details on page 9 lines

(194-196) as shown here- *'The IMFHL program was implemented in two phases: an initial trial or learning phase in limited areas from 2012 to 2015, followed by at-scale implementation from 2015 to 2017 (32,33). The program developed the final implementation design and scale-up strategy based on experiences in the pilot phase. At scale, the program implementation occurred from 2015 to 2017 in UP state and the survey data collected at the start and end of this period is used for this paper (32,33).'*

R1-5: Reviewer Comment : Additionally, a lot of the stuff written under this section, are not study design matters. For example, the stuff starting from “The maternal and neonatal health literacy component was layered upon a microfinance-based SHG platform towards 2015 to provide pregnant and recently delivered women with health messages in rural UP. A detailed description of the IMFHL program context, selection of intervention, comparison and control blocks is available elsewhere (32,33)..... This diffusion of knowledge from members

(tier I) to neighbouring non-members (tier II) is depicted in Figure 1 below”. iv. The authors are advised to remove this information and place it under a different subheading as “Description of the Intervention”.

Response: We thank the reviewer for these comments and have addressed comments in the paper as follows. We have moved the section and created a new subheading of '**Description**

of the IMFHL program intervention' on page 8, with the '**Methods'** section starting on page 9.

Sample size and sampling procedure

R1-6: Reviewer Comment :It is stated on page 9 that a detailed description of the selection strategy is available elsewhere (38,39). A brief description of the sampling procedure for households is still needed in this paper. Similarly, though details of the sample size determination are provided elsewhere, the reader of this paper needs to be informed of how this was done especially when he/she has no access to that information.

Response: We thank the reviewer for this query. We have added details on page 9 (lines 227-230), where the sampling procedure is mentioned and page 10 (lines238-241), where sampling details are now provided. The IMFHL program used a multistage stratified sampling process which comprised of a selection of block, GP (village) and household. Each sampling process has comprehensively been detailed in our other publications in open access journals, provided as references (38, 39-now 32,33).

Since this paper is not about the implementation strategy of the program and the program sampling methodology is common to other papers using the same IMFHL dataset, we feel that, providing the sampling details again in this paper will limit the scope of this study. We are rather highlighting the program impact on change in behaviour related to maternal health outcomes related to BPCR which is a novel finding.

Statistical analysis

R1-7: Reviewer Comment : *For the sake of order and logical flow of study activities, the **statistical analysis** section should come after the section on measurement of outcome, exposure, and confounding variables. What statistical package was used for data analysis? SPSS? Stata? etc.*

Response: We thank the reviewer for these comments and as suggested, we have changed the ordering on pages 10-13. The statistical analysis section now appears on page 13 (lines 305--317) after the sections on measurement of outcome, exposure, and confounding variables. Additionally, we have also added details of the software package (Stata 16) that was used for the analysis on page 13 (lines 316-317) as follows- '*All analyses were performed using Stata 16 (Statacorp, USA).*'

RESULTS

R1-8: Reviewer Comment: *The "Descriptive Statistics" should follow a major heading titled "RESULTS" i. It is also important to compare the socio-demographic characteristics of study participants at baseline before the substantive results of the study. Any baseline differences not accounted for will confound the outcome of the study.*

Response: Authors thank the reviewer for the suggestion and have made changes as suggested. The '**descriptive statistics section**' now follows the '**Results**' sub heading on page 14, line 318. We also agree that baseline comparability with endline is important. While the IMFHL program's multistage stratified sampling strategy collected data in both rounds from same villages, and we have checked for comparable data from members and non-members in table 2, we also separately checked and confirmed the comparability of key descriptive variables in round 1 and round 2 (enclosed here as annexure 1 table1).

R1-9: Reviewer Comment: *Table 2 is too crowded and confusing because different information types including socio-demographic characteristics, antenatal attendance, individual health and health system characteristics of the study participants are all contained in the table. The table should be split appropriately into two distinct tables under separate sub-headings as follows: socio-demographic characteristics of the sample, past obstetric data and medical history of mothers (e.g., antenatal*

attendance, place of delivery etc.). It will be informative to add a measure of association (e.g., chi-squared statistic) between these variables and type of household (non-member households and SHG) presented in Table 2.

Response: Thank you, Dr Saaka, for this question. As suggested, we have conducted measures of association (T-Test and Chi-square) for the descriptive variables in table 2 and added a column to the table to report P values. We have also added lines 501-503 on page 16 to highlight that table 2 provides descriptive and comparison of SHG and Non-Members using independent samples T-test and Chi-square test. Comparability of sampled groups of women across SHG membership and non-membership was confirmed for key variables as presented in table 2, and all variables were adjusted for the regression models (III and IV).

Furthermore, as separately suggested in query (R1-8), we also confirmed comparability of sampled groups between round 1 and 2 (results attached as annexure 1). The layout of the current table is developed to aid readability considering key information in one place and we have improved on previous iterations of similar tables we published elsewhere (ref. 32) using the IMFHL data.

Our selection of key confounders was based on maternal health literature and we had categorised them across levels using the social determinants of health (SDH), which has a demonstrated effect on health outcomes. The SDH* emphasise the effect of individual, household and broader societal level factors which have informed the table and subsequent modelling (I-IV). For example, model

III assesses the program effect on outcome adjusting for the individual and health system variables, while model IV includes the socio-demographic and area-level characteristics.

Reference

- *World Health Organisation 2021, Social Determinants of Health (internet) WHO website available https://www.who.int/health-topics/social-determinants-of-health#tab=tab_1

R1-10: Reviewer Comment: *I am wondering whether the program exposure characteristics should be presented in Table 2 since they are not comparable across the different households.*

Response: Thank you for the suggestion. We considered this option, and we feel that providing the information in one table improves overall readability and aids the reader in assessing descriptive estimates across member and non-member households considering the various program exposure levels. Moreover, as the paper currently contains five tables; the authors think that creating additional tables potentially impacts the readability and presentation of the overall information. However as shared in response R1-9, we have added chisquare and t test measures to table 2 and report on statistical significance. We have included each variable in the regression models III and IV which is able to provide adjusted effects holding each variable constant.

R1-11: Reviewer Comment: *The information presented in Tables 1 and 2 are also part of study results and I am therefore wondering why they were presented before the section titled "RESULTS". Is it supposed to suggest that only the information presented in tables 3-4 are the results of the study?*

Response: Thank you for the suggestion. We made changes to the layout as per the previous query (R1-8) so that the 'Results' section title on page 14 (line 318) now precedes tables 1 and 2 as well as tables 3,4 and 5, which constitute study results.

R1-12: Reviewer Comment: *How was additive effect of health literacy on BPCR isolated from that SHG households membership alone? For example, women in SHG households were 1.17 times more likely to have practised BPCR in their last pregnancy (OR=1.17, 95% CI: 1.07–1.29, $p<0.01$) than non-members in pure control villages. SHG members who received additional health literacy were 1.48 times more likely to practise BPCR (OR=1.48, 95% CI: 1.35–1.63, $p<0.01$) than non-members in pure control villages. These were generated from two separate regression models. What is not clear is what will happen when all models are combined into one? What will be the independent effect of all four exposures in the regression model?*

Response: Thank you for the question. We isolated the additive effect of health literacy on SHG's plus health in model I, where we analysed the effect of different IMFHL exposure levels in one model (details in table 3 page 20,21). As described under the 'statistical analysis section page 13 (lines 304-316) and subsequently shown in tables 3 and 4 the four

regression models allowed us to understand the independent effect of all exposure levels together (model I) and then understand how the results changed when we included an interaction term (model II) and confounders (models III and IV) on the outcome.

R1-13: Reviewer Comment: Interestingly, non-member women living in the same villages where either SHG or SHG plus health intervention was implemented were 0.89 times less likely to practise BPCR compared to non-member women in pure control villages (OR=0.89, 95% CI:0.82–0.97, $p<0.01$).

Response: Thank you for the observation. We explored the effect of the program on non-members within survey rounds in model II where an interaction term was included and where we saw that non-members gained in health outcomes in round 2, when they lived in villages with a SHG plus health intervention.

R1-14: It is also reported that overall, results showed that women in households interviewed in round II were almost 0.37 times less likely to have practised BPCR compared to households interviewed in round I given all the levels of program exposure (OR=0.37, 95% CI:0.35–0.39, $p<0.01$). Were these women in the programme or non-programme communities? I guess this apparent negative effect arose because different households were involved in the baseline and follow-up surveys. I think this is a major limitation of this intervention study.

Response: Thank you for the query. The program implementation and data collection follow a

DID strategy with a selection of different geographies based on program exposure so that

program contamination doesn't occur and the independent effect of the program could be

identified.

To start with, in model-1, the round variable considers all women interviewed in round

1(2015, program baseline) and round 2(2017, program endline) irrespective of program

affiliation. We independently assess the effect of program exposure using the IMFHL program

variables, which identifies four levels of household exposure. Our model 1, thus, provide results

of the effect of time through the round variable, independent of the effect of program exposure and model-2 provide the interaction of both on outcomes. In the model-1, round 2 is compared with round 1 that represents the secular trend overtime, and it is pertinent to highlight that women left to themselves independent of the program exposure are 0.37 percent less like to practice BPCR (OR 0.37) in round 2 than in round-1. However, in the same model-1 the independent effect of program exposure reveals, for example SHG member HH in village with

SHG only intervention (OR=1.17) and SHG plus health member HH in village with SHG *plus* health intervention (OR=1.48) are 17 percent and 48 percent more likely to practice BPCR. Therefore, we further studied the effect of round across each *subgroup* of women across the program exposure in model II by including an interaction term between round and the different levels of program exposure variables.

Essentially, we find that across the four program exposure levels, women who were in pure control villages (no SHG, no health) report worse BPCR outcome in round 2 (OR=0.25) compared to the same group of women in round 1.

While our other subgroups SHG, SHG plus health and non-members in SHG plus health villages report higher likelihood of practicing BPCR in round 2 compared to the same household type in round 1.

This means that the net negative effect of the survey round (OR 0.37) in model-1, is mainly coming from women in pure control villages who are doing worse over time. The background reasons why this may be happening need to be explored, given the geography and program setting. One plausible explanation related to the type of outcome we are assessing. Full BPCR is a set of eight steps and, as a behaviour, is unlike traditional maternal health behaviours such as antenatal care utilisation, health literacy etc. which otherwise are reported in rural Uttar Pradesh (India) in general to have a net positive effect over time. Thus, the negative effect of time is important to understand for our study population, which reports continued poor maternal health outcomes.

Moreover, from our interaction results in model II, we are able to attribute a diffusion effect as non-members in SHG plus health villages are doing better (greater BPCR) compared to the same group in round 1. Moreover, the IMFHL program is exerting a strong effect to improve levels of BPCR in women who received SHG plus health intervention or lived in these villages who otherwise would have reported worse BPCR over time-as seen from pure control women. We also explain the above with an illustration below to show the overall negative effect of time on BPCR observed for Pure control groups, but the positive effect of IMFHL program in select households exposure groups where BPCR practice improved in round 2

*R1-15: I have reservations for the interpretation of some of the results. For example, on page 18 it was stated “the results showed that non-members in program villages reported 1.73 times higher odds of BPCR practice in round II compared to the same household type in round I (OR=1.73, 95%CI: 1.46–2.05, p<0.01)”. These results were obtained from different groups of women at different times (baseline and follow-up surveys). Since it was not the same group of women that were followed to round II, it will be difficult to conclude that a diffusion effect occurred from SHG women receiving health literacy to non-member women in the same villages, with women in this group being almost 1.194 (=0.69*1.73) (95%CI:0.89-1.57) times more likely to practise BPCR. I am of the conviction that the observed effect could be due merely to secular trends or just group differences and not to programme effect. A better evidence could be obtained if the comparison was made with different women in different non - programme communities. It should be remembered that the authors have earlier stated that non-member women living in the same villages where either SHG or SHG plus health intervention was implemented were 0.89 times less likely to practise BPCR compared to non-member women in pure control villages (OR=0.89, 95% CI:0.82–0.97, p<0.01). So, is it right to attribute any diffusion effect to the intervention?*

Response: Thank you for this note. We have addressed a similar query in the previous response

(pls see R1-14). While it is correct that the same households were not followed in both rounds as

explained in the sampling section (page 9), the IMFHL's sampling strategy allowed us to assess the programs impact and potential secular effect across four different levels of household exposure. To assess and attribute the diffusion effect to the intervention, we compared changes in outcome among non-member households in both SHG, and SHG plus health villages in round 2 versus round 1, but we also compared these changes in areas where there was no program at all to highlight the secular effect. Therefore, we cannot assign the diffusion effect that occurred from SHG women receiving health literacy to non-member women in the same villages, with women in this group being almost 1.194 ($=0.69 \times 1.73$) (95%CI:0.89-1.57) times more likely to practise BPCR, as merely due to secular trends or just group differences. However, we are careful to discuss our diffusion findings in the discussion section, where we recognise the study limitations and interpretation of inferences in the absence of randomisation and longitudinal data.

R1-16: Reviewer Comment: *Are the odds ratios presented in Table 3 crude or adjusted?*

Response: Thank you for the query. Table 3 (models I and II) provides main effects of program exposure, survey round and associated interaction, on the outcome (unadjusted for the potential effects of confounders) while table 4 (models III and IV) provide results that are adjusted for confounders. In essence, odds ratios in Table 3 are not crude, but the effect of one factor is adjusted for the others. We have added details to clarify these details in the statistical analysis section in page 14 (lines 314-315) where the model details are provided as follows '*Thus Models I and II provide program effect, unadjusted for the confounders, while models III and IV provide adjusted effects.*'

R1-17: Reviewer Comment: *In Table 4, there are insignificant predictors of BPCR in the regression models. Usually, insignificant variables are removed from the analysis since their presence does not give the true measures of association of the independent variables. The authors should re-run the regression and remove the non-significant variables in order to have a better understanding of which factors really were independent predictors of BPCR.*

Response: We have noted the query. We have identified a comprehensive set of variables from the maternal health literature and thorough clinical knowledge and experience that are individually related to maternal health and health system utilisation. Most maternal health papers fails to provide a comprehensive set of covariates of maternal health

behaviour which limit the model interpretation and generalisation. We retain the model with both significant and insignificant variables to enhance transparency in reporting as the results are unbiased estimates where the effect of one factor is adjusted for the presence of other potential covariates in the model. Also, our aim is to provide an analytical rather than a predictive model. Moreover, our modelling process is informed by the stepwise inclusion of variables in blocks across models I-IV, where blocks of variables related to respective social determinants of health are included in models III and IV, which are then assessed using various criteria to assess model fit such as log-likelihood ratio and Akaike and Bayesian information criterion (AIC /BIC), which are reported for models III and IV. The use of multivariable logistic regression modelling as done for this paper allows reporting of the association of each added variable independently with the outcome holding other factors constant.

R1-18: Reviewer Comment: Four separate regression models were fitted to the data. The second model (Model II) included an interaction term (IMFHL intervention by survey round) to draw out the change over time in the effect of IMFHL program exposure. What is not certain is whether in Model III and the full model, Model IV the interaction term was included in the modelling since that is not clear from table 4.

Response: Thank you for the query. We include an interaction term in models III and IV and the modelling results provided in table 4 for both shows the effect of adding confounders on the interaction term and changes in the association of program exposure and outcome.

R1-19: Reviewer Comment: The coefficient of the interaction term (time x treatment) will give the effect of the interventions with time. The full model is thus expected to contain main independent variable which in this case is the IMFHL intervention, all the independent covariates and the interaction term. The presentation of the results needs to be improved so as to understand the effect of the intervention over time. I suggest a table like the following will provide clarity to the results.

Response: Thank you for kindly providing the suggestion, Dr Saaka. As clarified in the previous response (R1-18), both models III and the full model IV include the main exposure variables (IMFHL program exposure and survey round), the interaction term (survey

round * program exposure) and the full set of confounding variables. We have also clarified details in the revised paper on page 21 (lines 404-406) as show ‘*Table 4 shows model III and IV results where the effect of main exposure variable, IMFHL program exposure and survey round, and interaction term is assessed when confounders are added.*’ Additionally, we have revised the reference category in table 4 as suggested. However, as the tables show, both model III and IV results are all similar to what is suggested, except the Wald statistics, which is not needed since we have the p-value and 95% confidence intervals. Therefore, we retain the original layout for a concise presentation of results to aid readability.

R1-20: Reviewer Comment: *To compare the changes over time in BPCR in intervention and control communities, a better stronger analytical option could have been the difference-in-difference (DID) analysis.*

Response: Thank you for the query. Thank you for the query. The IMFHL program’s stratified sampling and modelling conducted were informed by the principles of DID analysis. The IMFHL program collected data from intervention and comparator region(s), which are similar in baseline characteristics (details published as ref 38,39). This allowed, in the evaluation process, to assess the changes in program villages over time due to the program intervention compared to non-program villages. The study design was guided by the World Bank Handbook on impact evaluation (Khandker et al.2010*) in constructing the program exposure variable that allowed assessing the program outcome across different levels of household exposure based on DID for comparability. More importantly, the coefficient of the interaction term (time*treatment) in our regression model is the DID estimate, obtained at the log-scale for Logistic regression models. We then transformed these coefficients back to the multiplicative scale by computing the odds ratios.

*Reference

*“Khandker, Shahidur R.; Koolwal, Gayatri B.; Samad, Hussain A.. 2010. Handbook on Impact Evaluation: Quantitative Methods and Practices. World Bank. © World Bank.
<https://openknowledge.worldbank.org/handle/10986/2693> License: CC BY 3.0 IGO.”*

DISCUSSION OF STUDY RESULTS

R1-21: Reviewer Comment: *A discussion on the evidence of the impact of SHG membership alone and when integrated with health literacy on BPCR practice will be well-informed and clearer if some of the data is reanalysed and presented appropriately.*

Response: Thank you Dr Saaka. We have provided clarifications and detailed responses above, where we share that the modelling and subsequent discussion provide clear evidence of the impact of SHG membership and the SHG membership plus health literacy intervention on the outcome. We have rechecked our analysis and conducted additional tests as per the queries (see R1-8). Moreover, the modelling we used in this paper is the DID analysis (as the reviewer suggested (R1-20) and was also separately publication using similar methodology giving details, provided as reference 32* (*Ahmad D, et.al . The Knowledge of Danger Signs of Obstetric Complications Among Women in Rural India : Evaluating an Integrated Microfinance and Health Literacy Program. BMC Pregnancy Childbirth 21, 79 (2021). <https://doi.org/10.1186/s12884-021-03563-5>*)

Reviewer #2 Dr. KM Mehta, University of California, San Francisco:

Comments to the Author:

This article is about a very important topic, that is, the combination of microfinance and health layering to achieve maternal and child health outcomes. This paper focuses on a particular outcome, namely uptake of birth preparedness at the time of delivery. Some clarifications to the presentation of the analyses performed and the interpretation of the results are warranted.

Response: We thank Dr Mehta for providing constructive comments and the recognition of the paper's potential contribution to the literature. We have addressed comments in the paper as follows.

Section 'Methods':

R2-1. **Reviewer Comment** -*“How does the cross sectional analysis align with the IMF HL program timing? See below comments, this seems like 2 serial cross sectional surveys?”*

Response: Thank you for the comment. The two rounds of the cross-sectional surveys align with the baseline program assessment (round 1) conducted before the health intervention in 2015, followed by the end-line assessment (round 2) in 2017 at the end of program implementation. The paper provides these details on page 9, lines 190-191 and lines 195-197 as shown: *“The study used cross sectional data from a quasi-experimental survey design, collected as part of the IMFHL projectprogram in two rounds (pre-post program implementation), capturing the program’s characteristics and other factors in the first round in 2015, followed by a second round of data collection two years into the program in 2017 (328,339)”*

R2-2. **Reviewer Comment** *“Is this a healthy layering upon existing microfinance intervention?”*

Response: We thank the reviewer for this comment and agree with them. The IMFHL program comprised of health literacy layering on pre-existing microfinance (self-help groups) in ‘intervention’ areas. While separately, in comparator areas, microfinance only (without health layering) was implemented. We prefer to address these villages as comparison areas since we also have a group of villages where no-microfinance was implemented, and we call those areas as pure control areas. The duration of the microfinance platform was similar in intervention and comparison areas, with only the exception of added health layering in intervention areas. References providing IMFHL program intervention details are also provided in page 9 (lines 193 ref 32 & 33). We have added details on page 9 line 201 to clarify that health literacy layering was based on pre-existing microfinance intervention.

R2-3. **Reviewer Comment** *Do you have self help groups with health layering and self-help groups without? Was a statistical comparison made between these two?”*

Response: Thank you for the query. The program separately implemented SHG's with health layering (intervention areas) and without (comparison areas) and the paper has used appropriate statistical models to assess the difference in outcomes between the two areas. Statistical details are provided on page 14 lines 306-318. Moreover, the paper provides details of the construction of the exposure variable, the IMFHL intervention, page 12 (lines 258-272), allowing us to evaluate the program's main effect and the interaction effect over time (round X program) on the practice of BPCR, that is, the magnitude of change in BPCR practice across levels of IMFHL program exposure: intervention (SHG plus health), comparison (SHG only) and pure control (no SHG, no health) households. The evaluation methods used in our paper has also been published recently (Ahmad et al 2021), provided as reference (32) in the current paper.

R2-4. Reviewer Comment *How was the diffusion effect evaluated? What is the underlying theory of change?*

Response: The paper evaluated the diffusion effect from members to non-members in program intervention and comparison areas in regression models 1 and 2 results as reported in results (pages 20, lines 376-386) and table 3,4 (page 20, 21). These models assessed change of outcome (BPCR practice) amongst member and non-member households in both intervention and comparison areas. Moreover, we constructed the IMFHL explanatory variable using four household groups (0-4) depending on level of program exposure as detailed in page 13 (lines 280-289). The diffusion effect was assessed by finding the change in outcome for households that were not SHG members (non-members) in *program villages* (both comparison and intervention areas) where either the SHG only or SHG plus the health intervention was implemented compared to the pure control groups. The underlying theory of change is expected to occur through a process of collective socialisation in which health literate SHG members serve as role models and help other non-members on maternal and child health-related health literacy around pregnancy and childbirth. We have included the following lines 215-219 in page 10 to clarify the process as shown. We have included the following lines 215-219 in page 10 to clarify the process as shown - *An underlying process of collective*

socialisation (32,27) is expected to explain the diffusion process in which health literate SHG members serve as role models to help non-members in their community adopt protective behaviours around pregnancy and childbirth for safer maternal outcomes.

R2-5. Reviewer Comment *Though this trial is non randomised, can you delete me 8 what are the main socio-demographic differences?*

Response: Thank you for the question, we understand this query relates to comparability of households in intervention and comparison areas. The IMFHL program implementation and subsequent evaluation process was designed to select households of similar socio-economic traits in the intervention, comparison and control groups. The program used a three-stage stratified sampling process to ensure that intervention, comparison and control areas/households were matched for *socio-demographic* characteristics and were similar. We provide references to the detailed matching and sampling process in page 10 (references 32,33). Moreover, in the current study, we provide descriptive statistics of members and non-members in table 2 (page 15) which shows similar socio-demographic characteristics.

R2-6. Reviewer Comment *What is comprised in the birth preparedness and complication readiness intervention?*

Response. The IMFHL program provided Birth preparedness and complication readiness intervention or BPCR information to women as per the steps outlined by the World Health Organization. The paper provides descriptive details of each step as table 1. The paper also describes steps in the outcome variable section in page 11 (lines 261-267) as shown: *'The outcome variable, a binary variable, was constructed with '0' representing partial or no BPCR preparation and '1' representing complete BPCR, defined as households that practiced all birth preparedness and complication readiness steps during the last pregnancy as outlined by the WHO.*

R2-7. Reviewer Comment *Once the health layering and the financed based assistance given to self-help group women at the same time?*

Response. Thank you for the query. In the IMFHL program, the community organisation started first with establishing SHGs in intervention and comparison areas. Subsequently for intervention areas, health layering and financial based assistance was given at the same time. We have provided details clarifying these details in page 9 (lines 201-202) and also provided references in line 204 where full implementation details are published (ref. 32,33).

R2-8. Reviewer Comment *P 9 of 40, top, it is unclear why 2015 data was used if data were collected from 2015-2017*

Response. Survey data was collected in the IMFHL program in two rounds 2015 and in 2017 to assess the pre and post-program implementation effect. Our study uses both rounds of data to evaluate program outcomes and provides details of study design on page 9, lines 190--193 as shown here: *'The study used data from a quasi-experimental survey design, collected as part of the IMFHL project in two rounds (pre-post program implementation), capturing the program's characteristics and other factors in the first round in 2015, followed by a second round of data collection two years into the program in 2017 (32,33).'*

R2-9. Reviewer Comment *Page 9 of 40 lines 15 through 20 this is a problematic definition of recipient, if I'm mother-in-law or sister-in-law who is given the information then no direct information is given to the one woman who needs the knowledge, please state this as a limitation*

Response: Authors thank the reviewer for this insightful comment. The pregnant woman received the health information directly from the SHG, whether she herself was a member or lived in a household where mother-in-law or sister-in-law was members, in which case the pregnant woman was invited to attend health literacy meetings. Studies show that health behaviours in rural India are shaped more by family members when health system services are weak, with

the mother-in-law playing an influential role. Thus, for some pregnant women, the SHG platform may have a broader role in involving other family members in the health literacy discussions along with pregnant woman which is an important consideration in rural UP where the mother-in-law is influential in guiding the adoption of health behaviors. This aspect is not necessarily a limitation of the study design but a feature of the SHG program being conducted in real-life settings. We have clarified this in the limitation section on page 30 lines 520-531 as shown here: *'The IMFHL program's definition of SHG membership which consisted of the pregnant woman herself or her mother-in-law or sister-in-law is an important consideration for understanding how the health literacy component informed the pregnant woman's health behaviour change in intervention areas. In intervention areas, the pregnant woman received health information directly from the SHG whether she was a member or lived in a household where the mother-in-law or sister-in-law was members. The pregnant woman was invited to attend health literacy meetings in the SHG. In the program, the majority (2/3rds) of the pregnant women was SHG members themselves, and (1/3rd) lived in a household where the mother-in-law or sister -in-law was the member. Thus, for some pregnant women, the SHG platform may have a broader role in involving other family members in the health literacy discussions. This is an important consideration in rural UP where family members, especially the mother-in-law, influence the adoption of maternal health behaviours.'*

R2-10. **Reviewer Comment** Page 9 of 40 lines 31 through 35 are these women incentivised to diffusing content or knowledge in their network.

Response: Thank you for the query. There was no incentives for knowledge diffusion to non-members. The diffusion process is linked to collective socialisation and change of community norms (social attitudes and cultural practices) influencing maternal health which has been mentioned on page 10, lines 213-215. The SHG platform is noted in

other publications (ref 36.37, page 10, lines 212-219) to improve social connectedness in communities by creating newer networks of women, which potentially enhance communication flow.

R2-11. Reviewer Comment *Page 11 of 40 lines one through 8 the statistical method described here including an interaction term to draw out the change overtime would suggest that the study design is a serial cross sectional study not a single cross sectional study.*

Response: Thank you for the suggestion. The paper provides details on page 9 (lines 190--

193) where details of the two separate cross sectional surveys are provided as part of the program's quasi-experimental survey design.

R2-12. Reviewer Comment *Page 11 of 14 lines 19 through 28 the outcome variable is reported for the last pregnancy what is a bit unclear is whether this last pregnancy would have occurred after the IMFHL intervention do you have specific birth date of the children to verify that the intervention took place prior to the birth?*

Response: Yes, that is correct Dr Mehta, the outcome variable was reported for last pregnancy that occurred in the 12 months prior to the survey. Women interviewed in round 1 (2015) would have delivered in the 12 months before the start of the program while those women interviewed in round 2 (2017) would have delivered after the health intervention was started. We have clarified these details on page 11 lines 239-242 as shown here: *The survey collected data from currently married women aged 15 to 49 years who had delivered an infant in the 12 months preceding the survey, and from the household head and village representatives (32, 33). Thus, eligible women interviewed in round 1 (2015) would have delivered in the 12 months before the program's start, while those women interviewed in round 2 (2017) would have delivered after the health intervention was started.*

R2-13. Reviewer Comment *Page 11 of 14 lines 38 through 49 did the BPCR training cover all key eight steps listed here was there any vet verification of knowledge acquisition?*

Response: Thank you for the question. The BPCR training in the IMFHL program covered all the eight steps listed in the paper. The IMFHL training design provided this health information through multiple strategies and channels that were appropriate to the cultural and social context of the study population. Vetting of knowledge acquisition was encouraged through program strategies that allowed the women to share their understanding and perceptions. Role models or women who adopted the behaviours or faced barriers would also feed their experience through meetings. We and IMFHL program colleagues have published details of program implementation elsewhere and provided references in the paper (ref. 32,33).

R2-14. **Reviewer Comment** *Page 13 of 14, lines 6-11, how were these different program exposures captured? By geography?*

Response: That's correct Dr Mehta; the IMFHL program used geography to initially implement the program across intervention, comparison and control areas, and subsequently, the data was collected and evaluated across different geographies comprising intervention and comparison areas. Our paper constructs the exposure variable to reflect a household's exposure to the IMFHL program across different levels that correspond to these geographies as shown on page 12-13, lines 280-289. We have also published the program implementation and evaluation design elsewhere and provide these as references (32,33) in the current paper.

R2-15. **Reviewer Comment** *Page 13 of 15 lines 43 through 47 it is unclear how models three and four are parameterised please give more details.*

Response: Thank you for the query. While models I and II assess the main exposure effect on the outcome, models III included set of confounders (or covariates) specific to individual maternal health status and health system, and model IV included additional socio-demographic and area-level variables. In essence, model III is model II plus individual-and health system related covariates while model IV is model III plus household and area levels covariates.

These are standard confounders identified from the maternal health literature that influence maternal health outcomes. Model III confounders for example, parity, previous pregnancy complications, number of ANC visits, etc. allowed us to assess the adjusted program effect on BPCR for these variables. Likewise, in the final model (IV) we added confounders such as household's socio-economic status, family type, education levels etc. to model III and assessed the adjusted program effect on BPCR.

R2-16. Reviewer Comment *Page 18 line 22, the respondents' educational level at 66% completing primary middle school up to year nine seems very high do you mean this is any educational level up to 9th standard?*

Response: Thank you for the question. In our sample, 66% corresponds to women who have received any level of education from primary/middle school (up to year 9) and above. Out of our sample, 34 % of women reported that they had no education, which corresponds to current literacy rates for women in rural Uttar Pradesh state (60 %).

R2-17. Reviewer Comment *Page 18 line 24 did you consider analysing no BPCR steps as the reference then having a category of incomplete BPCR and lastly, comparing completed BPCR steps practiced in pregnancy.*

Response: Thank you for the question Dr Mehta; this is a pertinent question. We considered an ordinal outcome variable as suggested and also ran an initial analysis. Partial knowledge reported high numbers which were expected as women would have done a few steps out of the eight recommended. However, from a maternal health outcome perspective, as all 8 steps are recommended by WHO for safer maternal health outcomes, we finally constructed our outcome variable as a binary variable highlighting those women who practised all steps as the positive event of interest and coded as 1. The binary construction of the outcome variable also improved the interpretation of the results considering the message the paper conveyed to the reader. Given that, if the woman is in the middle category (incomplete/partial BPCR) of the ordinal outcome variable, it is difficult to interpret their level of the BPCR practice requirements and the model would not produce meaningful interpretation. Moreover, it is difficult to interpret in an ordinal analysis

that will treat the women shift between completing one or two steps in BPCR practice same as the shift between 6 or 7 steps.

R2-18. Reviewer Comment *Page 19 line 38 the finding that non-member women living in shg villages for .89 times as likely to practice BPCR compared to non-member women in control villages this means that non-member met women in control villages actually had better BPCR why do the authors think this is the case were there any Co interventions that may have influenced BPCR in these communities?*

Response. Thank you for the interesting question. We considered this possibility and one plausible cause could have been that SHG and SHG plus Health were implemented in geographic areas with significant health and poverty issues (e.g., high maternal and infant mortality rates). Additionally, to answer reviewer's query on other co-intervention being run in pure control villages, the IMFHL program monitored the implementation of other health programs in their geography and reported no other programs were implemented at the same time. The change in BPCR among non-member women living in SHG and SHG plus health villages is assessed in both model I and model II (pages 20-22), and the results need to be considered across both models. Model 1 shows that non-member women tend to report lower BPCR practice than women living in pure control areas (no SHG and no health). We have ruled out differences in geographies as a probable reason as comparable areas were selected, and we found similar socio-demographic composition across areas. We also understand that women would have exposure to similar health system factors across program areas, and no known co-interventions were in place during the study period that may explain the difference. While we will further explore this question, we know from model II that non-member women living in villages with SHG plus health did report much higher BPCR uptake in round II than the same group of women in round I, which attests to the program's diffusion effect. This underlying significant interaction effect of program exposure and survey round exists as shown, from model II in table 3 and 4 (page 20, 23) and reported on pages (21,lines398-402) where we explain that non-members in program villages reported 1.73 times higher odds of BPCR practice in round II compared to

the same household type in the round I (OR=1.73, 95%CI: 1.46–2.05, $p<0.01$), reversing the odds for non-member women in SHG plus Health villages actually to have better BPCR practice than women in pure control villages in round II. This result indicates that a

diffusion effect occurred from SHG women receiving health literacy to non-member women in the same villages.

A detailed response has also been provided for reviewer 1's query 14 pn page 7-9 which also helps provide details to this query.

R2-19. Reviewer Comment *Page 19 blind 44 through 49 one reason that you may have found less practiced BPCR at follow-up or in round two compared to round one it is because of attrition it may be that those individuals who practised BPCR or differentially last to follow up please state this as a consideration in the discussion.*

Response: Thank you Dr Mehta, for this observation. We would also suggest referring to reviewer 1(R1-14)query on pages 7-9 where we address a similar question.

The initial observations in model I show lower BPCR practice probably because the underlying interaction effect is at play which is not evaluated in this model (both rounds combined). Moreover, in the IMFHL program, while the same villages, blocks and districts were visited in both survey rounds, different households (women) were sampled and interviewed in each round, as detailed on page 10 (lines 228-232). Random variation in sampling may occur between the rounds; however, the programs' stratified sampling design (published elsewhere ref 32,32) seeks to reduce variation and produces comparable samples across both rounds. Additionally, in both survey rounds (1 and 2), the IMFHL collected data from comparable number of pregnant women which further reduces the possibility of having less pregnant women in pure control villages in round II compared to round I in the survey study setting .

R2-20. Reviewer Comment *Page 19 end face odds ratio 0.96 is not 4% lower odds as the confidence bounds include one this odds can be interpreted as statistically similar.*

Response: Thank you Dr Mehta, we agree that the OR is not statistically significant at the usual significance level (0.05), and we have thus stated on the page 20, lines 379-380,

that the results are not statistically significant as shown here " ...round 1 with a statistically not significant four per cent lower odds of practising BPCR compared with non-member women in control villages in round 1 (OR=0.96, 95%CI: 0.84-1.11, $p>0.10$)."

R2-21. Reviewer Comment *Page 20 top lines four through 11 I find this description confusing please clarify*

Response: Thank you for the important comment. Model IV included standard confounders linked to health system utilisation that may impact the association between the IMFHL program and BPCR outcome. While maternal health literature usually assesses associations between four Antenatal care (ANC) visits received during pregnancy and health behaviours, we looked at 4 ANC visits in combination with key clinical tests that need to be done in each ANC visit as a marker for the quality of ANC received. Moreover, in our study population, rural UP, health system gaps in providing quality ANC, documented elsewhere (ref 39,40 in the paper), was an important consideration in the modelling. The regression results for model IV show that independent of the program effect, women who received at least 4 ANC visits during the pregnancy (along with key test done) still reported lower BPCR practice compared to those women who received less than four ANC visits and incomplete tests done. We expect that ANC visits may either focus on information and education counselling pregnant women, and when front line workers conduct additional tasks during ANC visits, the amount of overall counselling time is potentially reduced where information such as BPCR can be provided.

R2-22. Reviewer Comment *Page 20 middle paragraph I suggest creating it table that has data based on a similar analysis in round one and round two this will clarify what model you are comparing in this paragraph*

Response: Thank you for the suggestion. Table 3 on page 23 presents a comparison between Model-1 and 2. We consider adding the another table in model 2 results section which shows the interaction term results from model II. However, as table 3 has full model II in the next page on the paper, we feel adding an extra table here in the results section potentially duplicates the information and affected readability. We have

provided the IT(interaction term) table below for reference which we took from table 3 to highlight IT results only

Table : Logistic regression model II interaction term results estimating levels of BPCR: Odds ratios and associated 95% confidence intervals

	Explanatory variable name	Interaction Term effects
		Model II
		Odds ratio (95 % CI)
1.	Two-way interaction effects: Round # households (HH) microfinance (MF) Exposure	
	Round 1# Non-MF HH in pure control village	Reference
	Round 2 # Non-MF HH in a village with MF or MF plus health	1.73*** (1.46-2.05)
	Round2 # SHG member HH in village with MF program only	1.10 (0.90-1.33)
	Round2 # SHG plus health member HH in village with MF plus health intervention	2.21*** (1.82-2.68)

Note: Confidence intervals in parentheses; significant p-value shown as ***: p<0.01, **: p<0.05 and *: p<0.10.

Response: Thank you for the suggestion Dr Mehta. The value provided here (page 21, lines 400--403) is produced by multiplying the OR of model II for non-member women OR=0.69 (95%CI: 0.61-0.77) with the OR produced in the interaction results for the same women in model II, i.e., OR=1.73 (95%CI: 1.46-2.05). The final value produced OR=1.194 (95%CI: 0.89-1.57) with the interval containing 1, which is indicative of overlap of the null value suggesting statistical insignificance.

R2-24. **Reviewer Comment** *Page 20 2 model 3 I think the strongest finding in the paper is for round two SHG plus health member and this finding should be highlighted in the abstract.*

Response: We agree Dr Mehta, this is an important finding of the program's overall impact.

The abstract on page 2, lines -41-43 highlights this finding as shown: *"Furthermore, the*

odds of practising complete BPCR increase to almost fifty per cent when a maternal health literacy component is added to the SHGs. A diffusion effect was found for BPCR practice from SHG members to non-members when the health literacy component was integrated into the SHG model”.

R2-25. Reviewer Comment *Page 26 top I would be careful or cautious about the interpretation of the diffusion there may be many reasons that non SHG members in SHG plus health literacy villages have an increased effect perhaps it would be good for the authors to explore some examples.*

Response: Thank you for this important observation. We have cautiously stated in the discussion section page 27 and page 31 (lines 547-549-508) that additional research is required to ascertain the higher BPCR practice observed in non-SHG members in SHG plus health literacy villages. The IMFHL program has conducted social network analysis studies, separately referenced (36,37), and found that adding health to a microfinance platform does improve community cohesion and communication, which we too have explored. Ruducha et al (2018, 2019) previously conducted SNA studies in IMFHL program areas and, found that SHG platforms acted as a social glue to connect women in communities by increasing communication networks that previously did not exist. They found an increase in relationships within villages linked due to an increase in the density and number of connections in a village. In our study, we build on Ruducha et al.'s work and apply the diffusion theory to explain an increase in BPCR uptake in SHG plus health villages

Annexure 1

Supporting information for the response to reviewer 1 query 8

Table 1: Summary statistics of variables by survey rounds, round I baseline (2015) and round II end line (2017)

	Variable	Summary statistics (N=17,244)	
		Round I (n=8,723)	Round II (n=8,521)
Independent variables: Program exposure characteristics			
1.	Level of household (HH) microfinance (MF) exposure		
	0. HH in a village with no SHG, no health intervention (pure control – reference)	1,801 (21%)	1,908 (22%)
	1. Non-member HH in a village with SHG only program	1,918 (22%)	1,428 (17%)
	2. Non-member HH in a village with SHG plus health intervention	1,723 (20%)	1,307 (15%)
	3. SHG member HH in a village with SHG only intervention	1,713 (20%)	1,910 (23%)
	4. SHG plus health member HH in a village with SHG plus health intervention	1,556 (18%)	1,968 (23%)

Independent variables: Individual health and health system characteristics			
2.	Parity (number of previous pregnancies) of the eligible woman (EW)	Mean=2.4 (SD=1.43)	Mean=2.4 (SD=1.42)
3.	Any past pregnancy loss (due to spontaneous/induced abortion)		
	No previous pregnancy loss (reference)	6,753 (77%)	6,051 (71%)
	Previous pregnancy loss	1,970 (23%)	2,470 (29%)
4.	Any complication experienced in last pregnancy/labour or post-partum		
	No complication experienced (reference)	4,777 (55%)	4,246 (50%)
	Complication experienced	3,946 (45%)	4,275 (50%)
5.	EW's knowledge of the minimum number of antenatal check-ups (ANC) required during pregnancy		
	Incorrect knowledge (reference)	5,898 (68%)	4,869 (57%)
	Correct knowledge	2,825 (32%)	3,652 (43%)
6.	Received four or more ANC in last pregnancy (with urine/blood pressure /weight/abdominal/ultrasound tested in last ANC)		
	Not received (reference)	7,489	5,010

		Summary statistics (N=17,244)	
	Variable	Round I (n=8,723)	Round II (n=8,521)
		(85%)	(59%)
	Received four ANC with all tests done in last ANC	1,234 (14%)	3,511 (41%)
7.	Place of last delivery		
	Home delivery (reference)	1,804 (20%)	1,135 (13%)
	Institutional delivery	6,919 (79%)	7,386 (86%)
8.	Received three postnatal care visits (PNC) in the first seven days after delivery		
	Not received	8,023 (92%)	7,653 (89%)
	Received three PNC in first seven days after delivery	700 (8%)	868 (10%)
09.	Number of contacts with ASHA/ANM/AWW/SHG in last pregnancy	Mean=3.0 (SD=4.9)	Mean=5.65 (SD=4.53)
10.	Distance (kilometres) to primary health centre if not available in the village	Mean=5.3 (SD=5.12)	Mean=5.4 (SD=4.7)
	Independent variables: Socio-demographic/economic and area level characteristics		
11.	Village distance (kilometres) to closest town	Mean=1.28 (SD=0.61)	Mean=1.68 (SD=0.84)
12.	Population of village	Mean=5,518 (SD=6,011)	Mean=4,768 (SD=3,985)
13.	Household (HH) with Below Poverty Line (BPL)		

	Card		
	HH without BPL Card (reference)	5,374 (62%)	4,055 (48%)
	HH with BPL Card	3,349 (38%)	4,466 (52%)
14.	Household wealth quintile (poor to poorest)		
	1. Marginally Poor	1,373 (16%)	2,110(25%)
	2. Moderately Poor	1,429 (16%)	2,122 (25%)
	3. Poor	1,520 (17%)	1,945 (23%)
	4. Poorer	2,155 (25%)	1,270 (15%)
	5. Poorest	2,246 (26%)	1,074 (12%)
15.	EW Presently Working to earn cash, in kind or both		
	Not working	7,412 (85%)	6,960 (82%)
	Presently working	1,311 (15%)	1,561 (18%)
16.	Family type		
	Nuclear (reference)	4,004 (46%)	3,162 (38%)
	Joint and extended family	4,719 (54%)	5,359 (62%)
17.	Religion		
	Muslim (reference)	710 (8%)	707 (8%)

		Summary statistics (N=17,244)	
Variable		Round I (n=8,723)	Round II (n=8,521)
	Hinduism and others	8,013 (92%)	7,814 (92%)
18.	Scheduled caste		
	General caste	1,213 (14%)	910 (11%)
	Other backward caste	3,551 (41%)	3,883 (46%)
	Scheduled caste and scheduled tribe	3,959 (45%)	3,728 (43%)
19.	EW age in completed years	Mean=26 (SD=4.46)	Mean=26 (SD=4.67)
20.	EW's education level		
	No schooling (reference)	3,369 (39%)	2,397 (28%)
	Completed primary/middle school (up to Year 9) and above	5,354 (61%)	6,124 (72%)

Acronyms: Eligible woman (EW), Self-help group (SHG), Accredited Social Health Worker (ASHA)/Auxiliary Nurse Midwife (ANM) Anganwadi worker (AWW). ASHA/ANM & AWW are government health workers in villages as per population guidelines providing preventative maternal, child and other health services.

Note: Non-member households are those which do not include a SHG member and SHG households are those that include a SHG member.

VERSION 2 – REVIEW

REVIEWER	Saaka, Mahama University for Development Studies, Community Nutrition
REVIEW RETURNED	03-Dec-2021
GENERAL COMMENTS	The authors have addressed my earlier concerns adequately. I have no further queries.